# Category-Extensible Out-of-Distribution Detection via Hierarchical Context Descriptions

**Kai Liu**[1,2*], **Zhihang Fu**[2†], **Chao Chen**[2], **Sheng Jin**[2], **Ze Chen**[2],
**Mingyuan Tao**[2], **Rongxin Jiang**[1†], **Jieping Ye**[2]

[1]Zhejiang University,  [2]Alibaba Cloud

## Abstract

The key to OOD detection has two aspects: generalized feature representation and precise category description. Recently, vision-language models such as CLIP provide significant advances in both two issues, but constructing precise category descriptions is still in its infancy due to the absence of unseen categories. This work introduces two hierarchical contexts, namely perceptual context and spurious context, to carefully describe the precise category boundary through automatic prompt tuning. Specifically, perceptual contexts perceive the inter-category difference (e.g., cats vs apples) for current classification tasks, while spurious contexts further identify spurious (similar but exactly not) OOD samples for every single category (e.g., cats vs panthers, apples vs peaches). The two contexts hierarchically construct the precise description for a certain category, which is, first roughly classifying a sample to the predicted category and then delicately identifying whether it is truly an ID sample or actually OOD. Moreover, the precise descriptions for those categories within the vision-language framework present a novel application: CATegory-EXtensible OOD detection (CATEX). One can efficiently extend the set of recognizable categories by simply merging the hierarchical contexts learned under different sub-task settings. And extensive experiments are conducted to demonstrate CATEX's effectiveness, robustness, and category-extensibility. For instance, CATEX consistently surpasses the rivals by a large margin with several protocols on the challenging ImageNet-1K dataset. In addition, we offer new insights on how to efficiently scale up the prompt engineering in vision-language models to recognize thousands of object categories, as well as how to incorporate large language models (like GPT-3) to boost zero-shot applications.

## 1 Introduction

Out-of-distribution (OOD) detection focuses on determining whether an input image is in-distribution (ID) or OOD, while classifying the ID data into respective categories [17]. The key to OOD detection has two aspects: (1) constructing a sufficiently generalized feature representation capability, so that images of arbitrarily different categories can be roughly separated from each other regardless of semantic shifts [67]; and (2) acquiring precise descriptions (namely decision boundary) for each ID category in the image feature space (in CNNs the last fully-connected layer plays this role), so as to determine whether the input image belongs to the corresponding category or just OOD [22].

Specifically, generalized feature representation comes from large-scale and diverse training data for learning-based data-driven models. Therefore, OOD detection works[33, 13, 50] have made great progress on the basis of large-scale pre-trained models such as ResNet [19] and ViT [14]. To achieve

---

*Work done during Kai Liu's research internship at Alibaba Cloud. Email: kail@zju.edu.cn.

†Corresponding authors. Email: rongxinj@zju.edu.cn, zhihang.fzh@alibaba-inc.com.

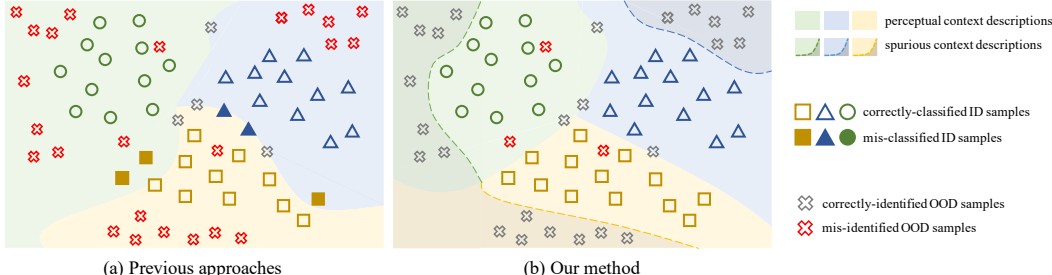

(a) Previous approaches          (b) Our method

Figure 1: **Method comparison.** Compared to previous approaches, our method utilizes the perceptual context to classify different categories under the current ID task (solid lines), and leverages the spurious context to strictly define the category boundaries independent of the current setting (dashed lines). The hierarchical perceptual and spurious contexts jointly describe the precise and universal boundaries for each category (combination of solid and dashed lines).

the precise description for a certain ID category, on the other hand, we need to let the model acquire as much "prior knowledge" as possible. For instance, if the description comes from a binary classifier that has only seen *cat* and *apple* during training, the description of category *cat* is obviously not precise enough to distinguish a *panther* image as an OOD: it lacks prior knowledge to distinguish different quadruped mammals.

Recently, vision-language models such as CLIP [41] provided significant advances in the two key issues above for OOD detection. Huge corpora of paired image-text data training brings both powerful feature representation capabilities and more comprehensive prior knowledge than single-vision-modal training [71, 37]. Therefore, CLIP-based OOD detection methods have been proposed successively, such as the zero-shot method MCM [38] and the encoder fine-tuning method NPOS [50]. However, there still exist the issues of generalization of feature representations and precision of category descriptions under the multi-modal framework.

Our observations can be summarized into two aspects: 1. The performance of the zero-shot CLIP-based OOD methods [38] is limited. As CLIP is trained by contrastive learning with informative image-text caption pairs, simply using the category names as all text information constrains the potential image-text discriminability. 2. Even though fine-tuning CLIP's encoder may boost the performance [18, 50], the generalization of multi-modal feature representation is sacrificed. In other words, such methods impair the model's ability to resist data shifts. Although the existing works are promising and inspiring to academia, constructing precise category descriptions via CLIP-like multi-modal features for OOD detection is still in its infancy [18].

Based on the observations above, we take a step towards generalized OOD detection by seeking the precise description prompt for each category and meanwhile maintaining the representation capacity. To this end, we develop a pair of perceptual context and spurious context for each in-distribution category to hierarchically construct the precise category description. CLIP encoders are kept frozen, and the contexts are learned through automatic prompt tuning.

As illustrated in Fig. 1, perceptual contexts first roughly classify a sample to the predicted category (colored regions), and spurious contexts then delicately identify whether it is truly an ID sample or actually spurious OOD (gray regions). Specifically, perceptual contexts perceive the descriptions of inter-category differences across all ID categories, while spurious contexts are relatively independent of the specific ID data task. The latter models a more strictly defined description of the current category itself by training on both real ID data and well-designed adversarial samples. We also introduce a robust sample synthesis strategy that leverages the prior knowledge of the VLM to select the qualified syntheses inspired by [50, 60].

We conduct extensive experiments to show the proposed hierarchical context descriptions are crucial to precisely and universally define each category. As a result, our method consistently outperforms the competitors on the large-scale OOD datasets, while showing comparable or even better generalization than the remarkable zero-shot methods. With the vision-language prompting framework, the precise and universal category descriptions via hierarchical contexts present a novel application: CATegory-EXtensible OOD detection (CATEX). We then merge the context descriptions learned from different

task data, and directly test on the union ID setting. The competitive results demonstrate that the learned descriptions can be used across tasks. Consequently, to illustrate the category-extensibility, we incrementally extend the context descriptions to the whole ImageNet-21K categories[11] at an acceptable GPU memory cost, and achieve superior performance to rivals. In addition, CATEX offers new insights on how to incorporate large language models (like ChatGPT) to boost zero-shot classifications (*e.g.*, implicitly constructing spurious contexts to perform one-class OOD detection).

In summary, we make the following contributions:

1. We construct the perceptual and spurious contexts to hierarchically describe each category to perform ID/OOD detection. Our method consistently surpasses the SOTA method on the challenging ImageNet-1K benchmark, leading to an 8.27% decrease in FPR95.

2. We empirically demonstrate the proposed hierarchical contexts are prone to learning precise category descriptions while keeping the generalized feature representation. When a data shift occurs, our method shows stable robustness on both ID classification and OOD tasks.

3. We present a novel category-extensible (CATEX) OOD detection framework. By simply merging the successively learned hierarchical contexts, we show that precise descriptions enable cross-task ID classification and OOD detection. We hope to offer new insights on how to scale up the prompt engineering in VLMs to recognize thousands of categories, as well as how to incorporate LLMs to boost zero-shot applications.

## 2 Preliminaries

This paper considers the multi-class classification as the in-distribution (ID) situation, and the OOD detection task is formalized as follows.

**Notations.** Let $\mathcal{X} \in \mathbb{R}^{H \times W \times 3}$ denote the input image space and $\mathcal{Y}^{in} = \{1, 2, \cdots, C\}$ denote the ID label space of $C$ categories in total. The labeled training set $\mathcal{D}_{tr}^{in} = \{(I_i, y_i)\}_{i=1}^{n}$ is drawn *i.i.d.* from the joint data distribution $\mathcal{P}_{\mathcal{X}\mathcal{Y}^{in}}$. Let $\mathcal{P}_{\mathcal{X}}$ denote the marginal distribution on $\mathcal{X}$, which is referred to as the in-distribution (ID). For ID classification task, let $f : \mathcal{X} \mapsto \mathbb{R}^C$ denote a function predicting the label of an input sample by $\hat{y}_i = argmax_k f[k](I_i)$.

**In-distribution classification.** In vision-language models, let $\mathbf{x} \in \mathbb{R}^d$ denote the $l_2$-normalized $d$-dimension visual feature of input images via the image encoder: $\mathbf{x} = \mathcal{I}(I)$, the mapping function $f$ can be expressed as $f(I) = W^T \mathcal{I}(I) = W^T \mathbf{x}$, where $W = [\mathbf{w}_1; \mathbf{w}_2; \cdots; \mathbf{w}_C]^T \in \mathbb{R}^{d \times C}$ is the collection of $l_2$-normalized text features for $C$ categories. Specifically, each text feature is extracted from the category description via text encoder: $\mathbf{w}_k = \mathcal{T}([v_{k,1}; v_{k,2}; \cdots; v_{k,m}; \text{CLS}_k]) \triangleq \mathcal{T}([\mathbf{v}_k; \text{CLS}_k])$, where $\mathbf{v}_k$ presents the $m$ word embedding vectors for $k$-th ID category and $\text{CLS}_k$ is the given category name. The ID-label prediction becomes $\hat{y} = argmax_k \mathbf{w}_k^T \mathbf{x} = argmax_k \langle \mathbf{w}_k, \mathbf{x} \rangle$.

**Out-of-distribution detection.** When an OOD sample $\mathbf{z}$ with unknown category $y \notin \mathcal{Y}^{in}$ emerges at test time, it should not be predicted to any known category. Hence, the OOD detection task can be viewed as a binary classification problem. Let $g : \mathcal{X} \mapsto \mathbb{R}^1$ denote a function distinguishing between the in- *v.s.* out-of-distribution samples. OOD detection estimates the lower level set $\mathcal{G} := \{\mathbf{z} : g(\mathbf{z}) \leq \lambda\}$, where samples with lower scores $g(\mathbf{z})$ are detected as OOD and vice versa. The threshold $\lambda$ is typically chosen by ensuring a high fraction of ID data (*e.g.*, 95%) is correctly preserved.

## 3 Method

From the data-distribution perspective, the key to OOD detection is to acquire precise descriptions for each ID category to determine the distribution boundary. Inputs beyond the specific boundary should belong to either OOD samples or other ID categories. However, it has always been a challenge [22, 17, 38]. In conventional methods using vision or multi-modal models, one may fine-tune the image encoder to reshape the feature representation space. Then, the classification layer (for vision models) or text features (for VLMs) play the role in category descriptions, where calculated logits measure the relative distances between inputs and corresponding ID categories. However, due to the absence of unseen categories, both the over-fitted feature representations [28] and the ID-label-biased category descriptions [4] are prone to overconfident predictions on unseen OOD data [40].

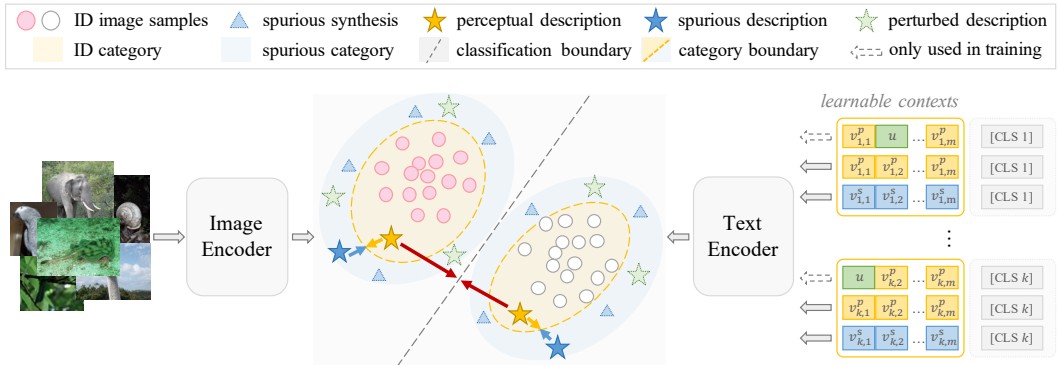

Figure 2: **Illustration of our method.** Perceptual context perceives a certain ID category, and spurious context explicitly describes a spurious category around this ID category. Random perturbation is applied to the perceptual context for synthesizing outliers to train the non-trivial spurious context. The hierarchical perceptual and spurious contexts jointly describe the precise category boundary.

To alleviate these problems, we take the vision-language prompting framework [71] to learn precise and universal descriptions for each ID category. First, both the image and text encoders are frozen to preserve the generalized representation capacity. Second, we propose to hierarchically construct the perceptual and spurious context for each category to mitigate label bias.

In the following sections, we present the proposed CATEX by elaborating on the following questions: 1. How do perceptual and spurious contexts model the precise and universal category descriptions (Sec. 3.1)? 2. How to learn such hierarchical contexts (Sec. 3.2)? 3. How to leverage the precise category descriptions for ID classification and OOD detection (Sec. 3.3)? The overview framework of our method is illustrated in Fig. 2.

## 3.1 Hierarchical Contexts

Recall that empirical risk minimization (ERM) [54] operates under the closed-world assumption. When an OOD sample $\mathbf{z} \notin \mathcal{P}_{\mathcal{X}}$ emerges, the ID classifier might produce overconfident predictions as $\hat{y} = argmax_k \langle \mathbf{w}_k^p, \mathbf{z} \rangle$, where $\mathbf{w}_k^p = \mathcal{T}([\mathbf{v}_k^p; \texttt{CLS}])$ is the encoded text feature for $k$-th ID category. Here $\mathbf{v}_k^p$ is denoted as **perceptual context** that perceives the original multi-category classifications. However, with $\mathbf{v}_k^p$ only, whether $\mathbf{z}$ actually belongs to the $k$-th ID category or is just an unseen OOD semantically close to this category is still unclear. Hence, we propose to add a hierarchical context to model such OOD samples as another spurious category. The image-text similarity is measured by $\langle \mathbf{w}_k^s, \mathbf{z} \rangle$, where $\mathbf{w}_k^s = \mathcal{T}([\mathbf{v}_k^s; \texttt{CLS}])$. Here $\mathbf{v}_k^s$ is denoted as **spurious context** that identifies such semantically similar but spurious OOD samples. We propose to utilize such two types of hierarchical contexts to jointly model the precise and universal boundaries for each category.

First, perceptual context serves to carefully adapt to the current ID classification task. As we propose to model a potentially spurious category for $C$ ID categories respectively, the total category number is equivalent to grows to $C + C = 2C$ in the scope. When an ID sample $(\mathbf{x}_i, y_i)$ emerges, to learn a more strict classification boundary, we put it apart from both other ID categories and their corresponding spurious categories by:

$$\underset{\mathbf{w}^p, \mathbf{w}^s}{argmin} \quad \frac{1}{n} \sum_i \left[ -\log \frac{e^{\langle \mathbf{w}_{y_i}^p, \mathbf{x}_i \rangle}}{e^{\langle \mathbf{w}_{y_i}^p, \mathbf{x}_i \rangle} + \sum_{k \neq y_i}^{C} e^{\langle \mathbf{w}_k^p, \mathbf{x}_i \rangle} + \sum_{k \neq y_i}^{C} e^{\langle \mathbf{w}_k^s, \mathbf{x}_i \rangle}} \right] \quad (1)$$

Then, spurious context is hierarchically combined to describe the specific category boundary to handle the OOD detection. To formulate the open-world situation, when an ID sample $\mathbf{x}$ or OOD sample $\mathbf{z}$ with the predicted category $\hat{y}$ emerges, the OOD detection risk can be expressed as:

$$R = \mathbb{E}_{\mathbf{x} \sim \mathcal{P}_{\mathcal{X}}} \left[ \mathbb{1}\{\langle \mathbf{w}_{\hat{y}}^p \mathbf{x} \rangle < \langle \mathbf{w}_{\hat{y}}^s, \mathbf{x} \rangle\} \right] + \mathbb{E}_{\mathbf{z} \nsim \mathcal{P}_{\mathcal{X}}} \left[ \mathbb{1}\{\langle \mathbf{w}_{\hat{y}}^p, \mathbf{z} \rangle > \langle \mathbf{w}_{\hat{y}}^s, \mathbf{z} \rangle\} \right] \quad (2)$$

For each ID category, if we can proactively draw the spurious OOD samples and their category-predictions as $\mathcal{D}_{tr}^s = \{(\tilde{\mathbf{z}}_i, \hat{y}_i)\}_{i=1}^{\tilde{n}}$, the risk in Eq. (2) can be empirically minimized by:

$$\underset{\mathbf{w}^p, \mathbf{w}^s}{argmin} \quad \frac{1}{n} \sum_i \left[ -\log \frac{e^{\langle \mathbf{w}_{\hat{y}_i}^p, \mathbf{x}_i \rangle}}{e^{\langle \mathbf{w}_{\hat{y}_i}^p, \mathbf{x}_i \rangle} + e^{\langle \mathbf{w}_{\hat{y}_i}^s, \mathbf{x}_i \rangle}} \right] + \frac{1}{\tilde{n}} \sum_i \left[ -\log \frac{e^{\langle \mathbf{w}_{\hat{y}_i}^s, \tilde{\mathbf{z}}_i \rangle}}{e^{\langle \mathbf{w}_{\hat{y}_i}^p, \tilde{\mathbf{z}}_i \rangle} + e^{\langle \mathbf{w}_{\hat{y}_i}^s, \tilde{\mathbf{z}}_i \rangle}} \right] \quad (3)$$

In short, based on Eq. (1), perceptual context first learns to model a more strict inter-class decision boundary. Hierarchically, combining with Eq. (3), perceptual context and spurious context jointly model the precise category boundary, which is the key to OOD detection. More details on the whole training procedure are provided in Appendix A.1.

Intuitively, how to draw the spurious training set $\mathcal{D}_{tr}^s$ is crucial. This paper presents a well-designed sampling strategy to learn the hierarchical contexts.

## 3.2 Perturbation Guidance

Generating adversarial data samples has been widely studied in recent years [39, 43, 72], among which feature-space sampling is proven more tractable [15, 50]. In general, previous approaches randomly synthesize spurious samples away from ID features' clustering centers (with low likelihood or large distance), and treat them as visual outliers. To improve the quality of spurious syntheses, we provide a perturbation-guided sampling strategy that leverages the prior knowledge of the pre-trained large-scale vision-language models.

In practice, an unseen spurious sample generally shares part of visual characteristics with the ID images, and the rest remains different [18]. Under the vision-language prompting framework, in the learned perceptual context of a certain ID category $\mathbf{v}_k^p = [v_{k,1}^p; v_{k,2}^p; \cdots; v_{k,m}^p])$, each word embedding $v_{k,j}^p$ actually describes a visual characteristic for this category [71]. According to the contrastive learning objective during the pre-training process on image-text pairs, if the text is perturbed, it should not be paired with the original image. Correspondingly, if the learned word embeddings are perturbed, the encoded text feature should also not describe images from the original ID category. Thus, as shown in Fig. 3, we can proactively leverage the perturbation to guide the spurious syntheses to boost context learning. Specifically, after randomly applying a pertur-

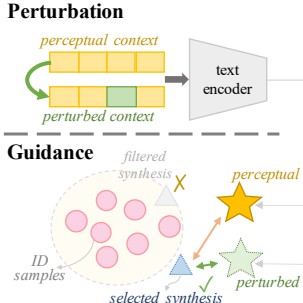

Figure 3: Guiding process.

bation $u$ (e.g., masking [29]) onto the word $v_{k,j}^p$ of perceptual context, the perturbed text feature becomes $\mathring{\mathbf{w}}_k^p = \mathcal{T}([v_{k,1}^p; \cdots; u; \cdots; v_{k,m}^p; \mathtt{CLS}_k])$. Rather than the original ID category, $\mathring{\mathbf{w}}_k^p$ now describes a kind of unknown spurious category, of which the images should share higher similarities with perturbed text feature $\mathring{\mathbf{w}}_k^p$ than the original $\mathbf{w}_k^p$. Hence, we propose to re-sample the randomly synthesized spurious candidates by:

$$\mathcal{D}_{tr}^s = \{(\tilde{\mathbf{z}}_i, \hat{y}_i) : \langle \mathring{\mathbf{w}}_{\hat{y}_i}^p, \tilde{\mathbf{z}}_i \rangle > \langle \mathbf{w}_{\hat{y}_i}^p, \tilde{\mathbf{z}}_i \rangle\} \quad (4)$$

Combining with Eq. (3) and Eq. (1), the spurious context $\mathbf{v}_k^s$ is able to learn to describe the synthesized spurious samples with such perturbation. After numerous iterations, random perturbations on perceptual contexts bring diverse spurious syntheses, which lead to learning non-trivial spurious contexts. More details on perturbation guidance (e.g., perturbed ratio) are discussed in Appendix A.2.

After learning the hierarchical perceptual and spurious contexts for each category, we are able to probe the precise and universal ID category boundaries.

## 3.3 Integrated Inference

After learning the hierarchical perceptual and spurious contexts, to further alleviate the overconfidence predictions on unseen OOD samples, we propose to regularize the vanilla image-text similarities by:

$$r_k = \langle \mathbf{w}_k^p, \mathbf{x} \rangle \times \frac{e^{\langle \mathbf{w}_k^p, \mathbf{x} \rangle}}{e^{\langle \mathbf{w}_k^p, \mathbf{x} \rangle} + e^{\langle \mathbf{w}_k^s, \mathbf{x} \rangle}} \triangleq s_k \times \gamma_k \quad (5)$$

Eq. (5) is a unified measurement. For ID classification, the category is predicted by $\hat{y} = argmax_k \, r_k$. For OOD detection, the commonly-used scoring function [38, 50] is adopted: $g(\mathbf{x}) = -max \frac{e^{r_k}}{\sum_j^C e^{r_j}}$.

The motivation is intuitive. When perceptual context $\mathbf{w}^p$ produces a relatively higher image-text similarity $s$ on unseen OOD samples or misclassified ID inputs, the hierarchical spurious context $\mathbf{w}^s$ could reduce the overconfident similarity by the regularization item $\gamma$. More empirical experiments are presented in the following sections.

# 4 Experiment

In this section, we empirically validate the effectiveness of our CATEX on real-word large-scale classification and OOD detection tasks. The setup is described below, based on which extensive experiments and analysis are displayed in Sec. 4.1-Sec. 4.2.

**Datasets.** Following the common benchmarks in the literature [59, 50, 60, 38], we mainly consider the large-scale ImageNet [11] as the in-distribution data. Subsets of iNaturalist [53], SUN [65], Places [69], and Texture [8] are adopted as the OOD datasets. The categories in each OOD dataset are disjoint with the ID dataset [25, 38, 50]. This paper investigates four practical scenarios in real-world application of ID classification and OOD detection tasks: (1) *standard OOD detection*, (2) *ID-shifted OOD detection*, (3) *category-extended ID classification and OOD detection*, and (4) *zero-shot ID classification*. Details are presented in Sec. 4.1.

**Evaluation metrics.** For OOD detection, two metrics are used: (1) FPR95, the false positive rate of OOD samples when the true positive rate of ID samples is 95%, and (2) AUROC, the area under the receiver operating characteristic curve. For ID classification, we report the mean accuracy (ACC).

**Implementation details.** Without loss of generality, our method is implemented based on the vision-language prompting framework [71] with the CLIP model [41], one of the most popular and publicly available pre-trained models. CLIP aligns an image and its corresponding textual description in the feature space through a contrastive objective. We mainly take CLIP-B/16 in our experiments, which comprises of a ViT-B/16 Transformer [14] as the image encoder and a masked self-attention Transformer [55] as the text encoder. During training, all parameters of the image and text encoders are frozen. We only learn a pair of perceptual and spurious contexts for each ID category. Following the default setting [71], each context consists of 16 learnable 512-D prompt embeddings, which are trained for 50 epochs using the SGD optimizer with a momentum of 0.9. The initial learning rate is 0.002, which is decayed by the cosine annealing rule. The contexts are optimized with synthesized samples, and no surrogate OOD datasets are involved during training.

## 4.1 Main Results

Table 1: OOD detection performance for ImageNet-1k [11] as ID dataset.

| Method | iNatualist | | SUN | | Places | | Texture | | Average | |
|---|---|---|---|---|---|---|---|---|---|---|
| | FPR95↓ | AUROC↑ | FPR95↓ | AUROC↑ | FPR95↓ | AUROC↑ | FPR95↓ | AUROC↑ | FPR95↓ | AUROC↑ |
| MCM [38] | 32.08 | 94.41 | 39.21 | 92.28 | 44.88 | 89.83 | 58.05 | 85.96 | 43.55 | 90.62 |
| MSP [18] | 54.05 | 87.43 | 73.37 | 78.03 | 72.98 | 78.03 | 68.85 | 79.06 | 67.31 | 80.64 |
| Energy [33] | 29.75 | 94.68 | 53.18 | 87.33 | 56.40 | 85.60 | 51.35 | 88.00 | 47.67 | 88.90 |
| ViM [57] | 32.19 | 93.16 | 54.01 | 87.19 | 60.67 | 83.75 | 53.94 | 87.18 | 50.20 | 87.82 |
| KNN [48] | 29.17 | 94.52 | 35.62 | 92.67 | **39.61** | **91.02** | 64.35 | 85.67 | 42.19 | 90.97 |
| SSD+ [45] | 59.60 | 85.54 | 75.62 | 73.80 | 83.60 | 68.11 | 39.40 | 82.40 | 64.55 | 77.46 |
| DOE [60] | 55.87 | 85.98 | 80.94 | 76.26 | 67.84 | 83.05 | 34.67 | 88.90 | 59.83 | 83.54 |
| VOS [15] | 31.65 | 94.53 | 43.03 | 91.92 | 41.62 | 90.23 | 56.67 | 86.74 | 43.24 | 90.86 |
| NPOS [50] | 16.58 | 96.19 | 43.77 | 90.44 | 45.27 | 89.44 | 46.12 | 88.80 | 37.93 | 91.22 |
| **Ours** | **10.18** | **97.88** | **33.87** | **92.83** | 41.43 | 90.48 | **33.17** | **92.73** | **29.66** | **93.48** |

**CATEX significantly improves *standard OOD detection*.** We first compare the proposed CATEX with competitive OOD detection methods that also delve into describing the ID category boundary. In particular, MCM [38] is the latest one to adopt the pre-trained CLIP to perform the ID boundary description in a zero-shot way, and others turn to fine-tune the encoders to optimize the ID boundary. The results are presented in Tab. 1, where the best performance is marked **bold**, and the

Table 2: Generalization across ID domains. Models are trained on ImageNet-1K [11] and directly tested on shifted ID datasets.

| Method | Target Datasets | | | | | | | | |
| | ImageNet-A | | | ImageNet-R | | | ImageNet-Sketch | | |
| | ACC↑ | FPR95↓ | AUROC↑ | ACC↑ | FPR95↓ | AUROC↑ | ACC↑ | FPR95↓ | AUROC↑ |
|---|---|---|---|---|---|---|---|---|---|
| MCM [38] | 50.61 | 71.52 | 80.65 | 75.95 | 52.67 | 89.49 | 47.42 | **73.73** | **80.74** |
| MSP [18] | 43.96 | 86.25 | 63.25 | 66.50 | 82.21 | 75.07 | 45.18 | 91.67 | 58.45 |
| Energy [33] | 43.96 | 89.78 | 63.96 | 66.50 | 89.55 | 68.43 | 45.18 | 95.56 | 54.32 |
| VOS [15] | 43.79 | 80.03 | 73.45 | 66.83 | 78.42 | 80.09 | 46.06 | 86.68 | 68.58 |
| NPOS [50] | 50.16 | 74.57 | 75.37 | 73.58 | 75.64 | 82.71 | **50.29** | 82.87 | 71.56 |
| **Ours** | **50.87** | **71.13** | **81.04** | **76.67** | **51.75** | **89.75** | 48.59 | 74.68 | 80.69 |

Table 3: Generalization across ID tasks. Models are independently trained on disjoint ImageNet-100 (I) and ImageNet-100 (II), and then tested on the union ImageNet-200 (I ∪ II) without fine-tuning.

| Method | ID Datasets | | | | | | | | |
| | ImageNet-100 (I) | | | ImageNet-100 (II) | | | ImageNet-200 (I ∪ II) | | |
| | ACC↑ | FPR95↓ | AUROC↑ | ACC↑ | FPR95↓ | AUROC↑ | ACC↑ | FPR95↓ | AUROC↑ |
|---|---|---|---|---|---|---|---|---|---|
| MCM [38] | 89.00 | 24.38 | 95.59 | 90.48 | 19.84 | 96.44 | 83.35 | 27.18 | 94.88 |
| MSP [18] | **94.70** | 41.34 | 93.39 | **94.82** | 38.22 | 93.75 | 87.69 | 60.93 | 87.26 |
| Energy [33] | **94.70** | 32.11 | 94.36 | **94.82** | 32.95 | 93.86 | 87.69 | 37.78 | 92.62 |
| VOS [15] | 94.68 | 25.19 | 95.60 | 94.72 | 20.97 | 96.01 | 86.14 | 34.16 | 93.42 |
| NPOS [50] | 94.24 (+0.00) | 16.54 (-0.00) | 96.62 | 94.32 (+0.00) | 16.84 (-0.00) | 96.35 | 86.23 (+0.00) | 25.54 (-0.00) | 94.35 |
| **Ours** | 94.12 (-0.12) | **10.31** (-6.23) | **97.82** | 94.42 (+0.10) | **7.91** (-8.93) | **98.31** | **89.61** (+3.38) | **13.34** (-12.20) | **97.19** |

second best is underlined. Accordingly, our CATEX achieves superior OOD detection performance, outperforming competitive rivals by a large margin. Specifically, CATEX consistently surpass the SOTA method NPOS [50] on all of the four OOD datasets, leading to an 8.27% decrease in FPR95 and 2.24% increase in AUROC on average. It implies that compared to zero-shot probing and encoder-fine-tuning, learning the hierarchical perceptual and spurious contexts for each category is more effective in acquiring precise ID boundaries. In addition, our method dose not conflict with other post-hoc approaches (such as ReAct [46] and ASH [12]), and appropriate combinations may bring further improvements (*e.g.*, **27.56%** of FPR95), as discussed in Appendix B.2.

**CATEX properly generalizes to *ID-shifted OOD detection*.** If the description of each category is precise enough, it will not only perform well in OOD tasks, but also be discriminative even when the distribution of ID data is shifted. Hence, we evaluate both the ID classification and OOD detection ability of models trained on ImageNet-1K [11] while tested on shifted ID datasets including ImageNet-A [20], ImageNet-R [23], and ImageNet-Sketch [56]. As shown in Tab. 2, CATEX reaches the best ID and OOD performance on both ImageNet-A and ImageNet-R and competitive results on ImageNet-Sketch. It indicates that the generalization ability from the large pre-trained models is well-maintained and even straightened. The fine-tuned models like NPOS are unstable across different datasets, which may be a sign of generalization degradation caused by parameter fine-tuning. Because the data shift is inevitable in real-world applications, we suggest that the proposed hierarchical contexts are instructive for model generalization.

**CATEX carefully adapts to *Category-extended ID classification and OOD detection*.** If the hierarchical contexts learn a precise category description, it should prevent overconfident predictions on any unseen samples in open-world but beyond the current ID task scope. We thus conduct a cross-task experiment to further evaluate the open-set discriminability. Specifically, two models are independently trained on ImageNet-100 (I) [38] and another disjoint ImageNet-100 (II) randomly selected from ImageNet-1K. Then, models are directly tested on the union ImageNet-200 (I ∪ II). OOD detection is simultaneously evaluated, and the results are displayed in Tab. 3.

Accordingly, our CATEX achieves the highest performance on the union ImageNet-200 and significantly surpasses the competitors (*e.g.*, 3.38% increase on accuracy and 12.2% decrease on FPR95). In the category-extended scenario, our classification accuracy only drops by 4%, which is much lower than the fine-tuning methods (*e.g.*, 8% decrease from VOS [15]). It implies that fine-tuning the encoder is prone to overfitting on seen training samples, and sacrificing the ability (obtained from

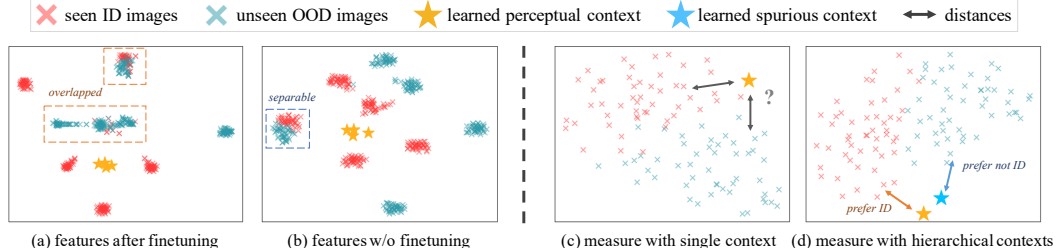

Figure 4: **Feature visualization by t-SNE**. (a) Previous approaches that fine-tune the encoders may distort the generalized feature space and make unseen OOD samples inseparable; (b) instead, our method freezes the encoders to maintain the discriminability. (c) Compared with traditional prompting methods using a single perceptual context only, (d) our spurious context provides a better metric for unseen OOD detection.

large-scale pre-training) to distinguish unseen categories. We randomly select five categories from each ImageNet-100 subset respectively, whose feature distribution shown in Fig. 4 is consistent with Tab. 3. Moreover, compared to the SOTA method NPOS, our CATEX outperforms more on the union ImageNet-200 than separated ImageNet-100 subsets (*e.g.*, 12.2% v.s. 6.23% decrease on FPR95). It further demonstrates that in each ImageNet-100 subset, besides our learned perceptual contexts that classify the current 100 categories, the spurious contexts are able to identify the unseen categories (both of the new-coming ID samples and OOD samples), as illustrated in Fig. 4. In this way, the precise boundaries for each category are acquired. On the other hand, to deal with such category-extensible tasks in real-world practice [10, 62, 61], this paper provides a new perspective: separately learning the precise descriptions for the new-coming categories besides the existing ones, and jointly testing on the full ID category scope.

To further evaluate the efficacy of the precise descriptions, we conduct a more challenging larger-scale category-extensible experiments: scaling up ImageNet-1K to ImageNet-21K [11]. In particular, we randomly split the huge ImageNet-21K dataset into several subsets, individually train the hierarchical perceptual and spurious contexts for each category on each subset, and concurrently test on the full ImageNet-21K. Such paradigm is similar to category-incremental learning, and the results are shown in Fig. 5, where we mainly compare with zero-shot CLIP [41] and baseline CoOp [71]. According to Fig. 5, our CATEX consistently outperforms CLIP and CoOp. Specifically, when the total number of categories extends, CoOp has a higher accuracy drop, since its learned contexts on each separated subset share limited discriminability to novel categories across

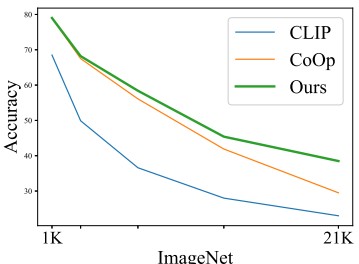

Figure 5: Scaling up ImageNet-1K [11] to ImageNet-21K [11] with category-incremental learning.

subsets. And training on twenty-thousand categories together is challenging due to the generally prohibitive computational costs, which require over *300 GB* GPU memory for the text-encoder on 21K categories. To alleviate this problem, this paper proposes to learn the hierarchical contexts to describe the precise and universal category boundaries, and achieves 38% accuracy on the full ImageNet-21K with a single V100-32G GPU card. We hope to offer new insights to the community on how to adopt large-scale VLMs to classify numerous categories with limited GPU resources.

**CATEX encouragingly boosts *zero-shot ID classification*.** Under the category-extended scenario, the hierarchical contexts help establish the category boundary for each given category, and prevent overconfidence on other unseen categories by adjusting the image-text similarity $s$ with the spurious-context-regularized item $\gamma$, as introduced in Eq. (5). Intuitively, in zero-shot classification scenarios, the regularized score $r = s \times \gamma$ may also rectify the category predictions and lead to a higher classification accuracy, with proper perceptual and spurious contexts. To verify it, as CLIP's default prompt template "A photo of a [CLS]" contains no visual information, we adopt rich visual descriptions from large language models [37] (such as GPT-3 [3]) as our perceptual context $\mathbf{w}_p$ to perform classification (denoted as *VisDesc*). Then, we randomly perturb the visual description to simulate spurious contexts $\mathbf{w}_s$, and regularize text-image similarities via Eq. (5). The results shown in Tab. 4 indicate the regularized score $r$ brings a higher classification accuracy, without any training

Table 4: Zero-shot classification via text-image similarity regularized by simulated spurious contexts.

| Method | Prompt Example | ACC-Top1↑ | ACC-Top5↑ |
|---|---|---|---|
| CLIP | A photo of a goldfish. | 63.50 | 88.99 |
| VisDesc | A yellow fish with a long flowing tail is goldfish. | 65.47 | 90.14 |
| Ours | + A [MASK] fish with a long [MASK] [MASK] is goldfish. | **65.84** | **90.36** |

cost. We hope to provide new insights that in the category-extensible setting or a true zero-shot classification scenario, explicitly constructing spurious contexts can perform the one-class OOD detection task, as other categories can also be viewed as OOD for a certain ID category.

## 4.2 Ablation Studies

In this section, we take ImageNet-100 [38] as the ID dataset, and other settings follow Sec. 4.1. We mainly verify the key contributions of this work in Tab. 5, and the influence on model capacity in Tab. 6. In particular, we evaluate the number of spurious contexts used to help probe the category boundary in Tab. 7. More experiments and discussions are displayed in Appendix A and Appendix B.

Table 5: Ablation study on proposed framework.

| Method | ID-ACC ↑ | FPR95↓ | AUROC↑ |
|---|---|---|---|
| baseline [71] | **94.12** | 13.07 | 97.42 |
| +Hierarchical-Contexts | 93.84 | 11.39 | 97.66 |
| +Perturbation-Guidance | 94.10 | 10.59 | 97.81 |
| +Integrated-Inference | **94.12** | **10.31** | **97.82** |

Table 6: Ablation study on model capacity.

| Model | Method | ID-ACC ↑ | FPR95↓ | AUROC↑ |
|---|---|---|---|---|
| **RN50** | MCM | 84.26 | 32.31 | 94.61 |
| | Ours | **90.58** | **19.58** | **96.25** |
| **ViT-L/14** | MCM | 91.66 | 20.79 | 96.35 |
| | Ours | **96.18** | **6.97** | **98.43** |

**The proposed components are effective**. In Tab. 5, a baseline model is first constructed following the vision-language prompt-learning framework CoOp [71], which only learns the perceptual contexts with the vanilla softmax cross-entropy loss. Then, we learn the hierarchical perceptual and spurious contexts using initially synthesized OOD samples with Eq. (1) and Eq. (3). According to Tab. 5, the OOD detection performance is improved immediately, whereas the ID classification accuracy slightly drops by 0.3%. It indicates that the quality of initial OOD syntheses is limited, which puts the learned descriptions at risk of identifying ID inputs as OOD or even other ID categories. On the contrary, the proposed perturbation guidance mechanism in Eq. (4) provides extra constraints on OOD syntheses to improve the quality, which brings consistent gains in both ID classification and OOD detection protocol. Finally, the integrated inference strategy in Eq. (5) brings free improvement on FPR95 without loss of ID accuracy. The efficacy of our proposed method is further demonstrated.

**Our method is scalable to model capacity**. In Tab. 6, we investigate the scalability of our method with different model capacities, *i.e.*, a lighter model with modified ResNet50 (RN50) and a heavier model with ViT-L/14 as the image encoder. The results imply larger models indeed lead to superior performance, suggesting larger models are endowed with better representation qualities [38, 50]. Interestingly, though the zero-shot method MCM [38] with ViT-L/14 surpasses our method with RN50 by 1.1% of ID classification accuracy, the OOD performance is inferior to ours (*e.g.*, over 1.2% increase on FPR95). It further demonstrates that learning precise category descriptions via our hierarchical contexts is still a key to deriving a better OOD detector in the open-world.

**Multiple spurious contexts bring more precise descriptions**. For each ID category (*e.g.*, cat), the spurious OOD samples can be diverse (*e.g.*, panthers, lions, *etc*). Thus, we aim to only describe the spurious sample surrounding the ID category, rather than the whole OOD space (which is too complicated). According to common sense, it is intuitive to use more spurious contexts to describe better category boundaries. To verify it, we have tested the number of spurious contexts for each ID category (denoted as *W/o Constraints*), and the results shown in Tab. 7 implies using more spurious contexts only leads to 0.1% gain on performance. The reason may be that the learned 2 or more spurious contexts are too redundant without any constraints. To alleviate this problem, we simply add an orthogonal constraint (making the similarities between each two spurious contexts close to zero) (denoted as *W/ Constraints*), and the OOD detection performance is significantly boosted, as displayed in Tab. 7. Therefore, how to effectively and efficiently leverage more spurious contexts to better describe the category boundary deserves further exploration, and we view it as our future work.

Table 7: Ablation study on multiple spurious contexts.

| $N_s$ | W/o Constraints | | W/ Constraints | |
|---|---|---|---|---|
| | FPR95↓ | AUROC↑ | FPR95↓ | AUROC↑ |
| 1 | 10.31 | 97.82 | 10.31 | 97.82 |
| 2 | **10.21** | 97.86 | 10.17 | 97.86 |
| 4 | 10.27 | **97.88** | 9.89 | **97.89** |
| 8 | 10.25 | 97.86 | **9.76** | 97.84 |

## 5  Related Works

**OOD detection in computer vision.** To keep the safe deployment of multi-category classification models in open-world scenarios, the goal of OOD detection is to derive a binary ID-*v.s.*-OOD detector for the visual inputs. To reduce the overconfidence on unseen OOD samples [1], a surge of post-hoc scoring functions has been devised based on various information, including output confidence [21, 32, 25], free energy [33, 58, 15], Bayesian inference [30, 34, 5], gradient information [24], model/data sparsity [46, 73, 12], and visual distance [31, 49, 50]. Another promising approach is adding open-set regularization in the training time [36, 22, 26, 52, 63], making models produce lower confidence or higher energy on OOD data. Manually-collected [22, 60] or synthesized outliers [15, 50] are required for auxiliary constraints. Works insofar have mostly focused on regularizing the task-specific classification models trained on downstream scenarios using only visual information. In contrast, this paper explores the OOD detection on category-extensible scenarios, which incorporates rich textual information to probe the task-agnostic category boundaries.

**OOD detection with vision-language models (LVMs).** Exploring textual information for visual OOD detection with large-scale pre-trained models has recently attracted a surge of interest. One may simply use the vision-language model CLIP [41] to explicitly collect potential OOD labels [18, 16] or conduct zero-shot OOD detection [38] with manually pre-defined text prompts. However, as such manual text prompts lack the description of unseen categories, the prior knowledge learned by VLMs during pre-training is not fully exploited, leading to inferior OOD detection performance. On the contrary, we propose to learn hierarchical contexts to describe the category boundaries against unseen spurious categories via vision-language prompt tuning [71]. RPL [7] and ARPL [6] explored a similar idea of "Reciprocal Points" to constrain a more compact in-distribution feature space, which may distort the generalized representation like NPOS [50]. Our method largely boosts the OOD detection performance, and simultaneously maintains the generalization capacity of pre-trained VLMs.

## 6  Conclusion

This paper proposes a novel framework to learn the precise category boundaries via the hierarchical perceptual and spurious contexts. Specifically, the perceptual context first perceives a relatively more strict classification boundary on the current ID task, and then integrates the spurious context to determine a more precise category boundary that is independent of the current task setting. The extensive empirical experiments demonstrate the proposed hierarchical contexts are prone to learning precise category descriptions, which shows robustness to various data shifts. By simply merging the task-independent precise category descriptions, we provide a new perspective to efficiently extend the ID classification scope to twenty-thousand categories at an acceptable resource cost. We also offers new insights to boost zero-shot classification via text-image similarities regularized from an one-class OOD detection perspective.

**Limitations and societal impact.** To keep the representation capacity of visual features and the generalization ability to identify unseen samples, we freeze the image encoder of the pre-trained CLIP. It limits the upper-bound performance on ID classification and OOD detection of our method. Larger models (*e.g.*, ViT-L/14@336px) might bring higher performance but require more costs. There is a trade-off between efficacy and efficiency. Besides, as we do not fine-tune the encoder, the social biases accumulated during pre-training on uncurated web-scale datasets are inherited. Directly applying our method may cause biased predictions containing stereotypes or racist content. Leveraging debiasing techniques could be the key to alleviating this problem.

## Acknowledgments and Disclosure of Funding

This work was supported in part by the Fundamental Research Funds for the Central Universities, and in part by the National Key R&D Program of China under Grant 2020AAA0103902.

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

# Appendices

## A    Implementation Details

### A.1    Training Procedure

This section provides a walkthrough example to illustrate the training procedure presented in Sec. 3.1 of the manuscript. As displayed in Fig. A1, we first generate spurious OOD syntheses by perturbation guidance in Sec. 3.2, and then learn in-distribution classification via Eq. (1) and out-of-distribution detection via Eq. (3). Specifically, Fig. A1 captures the training procedure for one category. For the multi-category scenario, please refer to Fig. 2 in the manuscript.

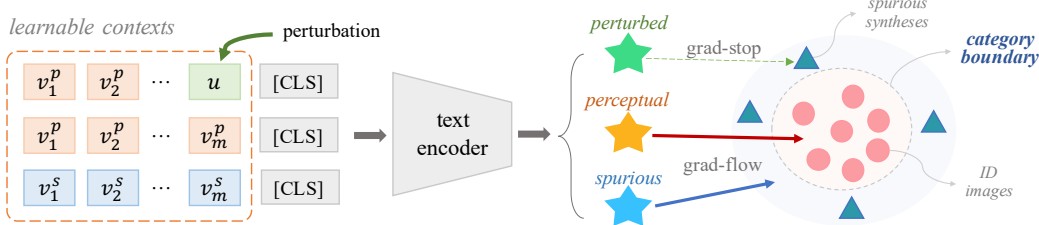

Figure A1: **Training illustration for one category.** The randomly perturbed context is adopted to select spurious OOD syntheses, where gradients are stopped. Then perceptual context describes the ID images, and spurious context describes those spurious OOD syntheses surrounding the ID category. The two contexts jointly model the category boundary.

### A.2    Perturbation Guidance

**Detailed perturbation guidance process**. For arbitrary $k$-th ID category, we use the spurious context to explicitly describe a corresponding spurious category, and a critical consideration is how to synthesize training samples spurious to that $k$-th ID category. Recently, generating adversarial data samples have been widely studied, including GAN networks [39, 27], diffusion models [43, 44], image attacks [35, 72], and feature-space sampling [15, 50]. For simplicity, we take the tractable feature-space sampling as NPOS [50] to generate spurious candidates. We calculate the $k$-NN distance for each ID sample in the specific category, and sample OOD candidates from a multivariate Gaussian distribution around the ID data points with largest distances (basically away from the clustering center). Then, we leverage the perturbed descriptions of perceptual context to guide the further filtering for high-quality spurious syntheses. Visualization examples are provided in Fig. A2.

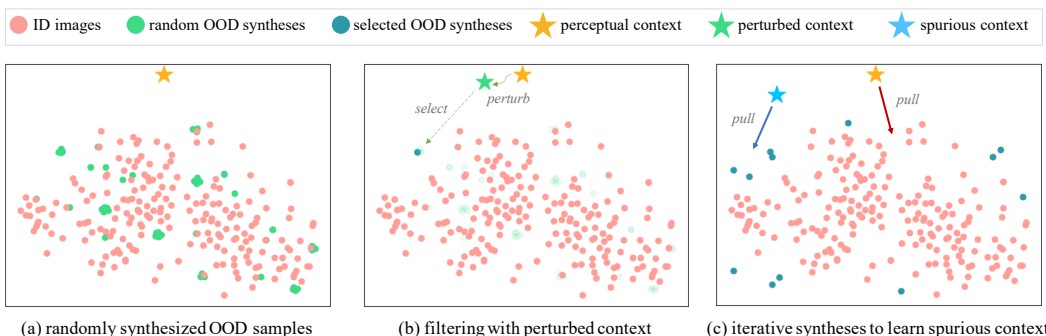

Figure A2: **Visualization of the Perturbation Guidance process.** (a) Given a set of in-distribution image samples, several OOD syntheses are randomly generated in the feature space. (b) We perturb the perceptual context to simulate a kind of OOD category, and select OOD syntheses subject to (more similar to) the perturbed context. (c) After iteratively sample generation and perturbation-guided selection, the synthesized OOD samples are adopted to learn the spurious context.

**Perturbation approach**. Given the perceptual context $\mathbf{v}_k^p = [v_{k,1}^p; v_{k,2}^p; \cdots; v_{k,m}^p])$ of $k$-th ID category, we randomly apply a perturbation $u$ onto one arbitrary $v_{k,j}^p$ to produce a perturbed description $\mathring{\mathbf{w}}_k^p$ through text-encoder. Intuitively, there are two ways to perturb a context $\mathbf{v}_k^p$: erasing or replacing the specific visual character $v_{k,j}^p$. Consequently, we design three types of perturbation: (1) masking with a placeholder $u = [\texttt{MASK}]$, (2) noise from a Gaussian distribution $u = \sigma$, and (3) swapping with another category $u = v_{k',j'}^p$. And the perturbed text-feature is produced by: $\mathring{\mathbf{w}}_k^p = \mathcal{T}([v_{k,1}^p; \cdots; u; \cdots; v_{k,m}^p; \texttt{CLS}_k])$. As shown in Fig. A3, all of the three types of perturbed text-features $\mathring{\mathbf{w}}_k^p$ slightly deviate from the original $\mathbf{w}_k^p$ while keep the affinity (*e.g.*, shares a 97% similarity against the original one.) Specifically, the noised $\mathring{\mathbf{w}}_k^p$ leads to a greater deviation, since the noised visual character $v_{k,j}^p := u$ is more unpredictable than the masked or swapped ones.

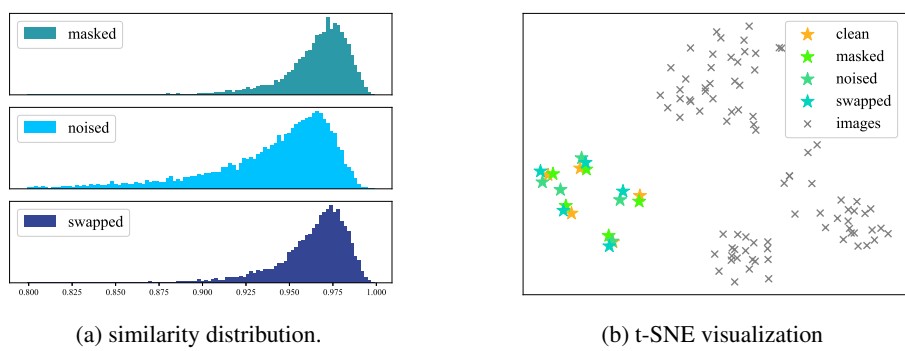

(a) similarity distribution.      (b) t-SNE visualization

Figure A3: **Statistics of perturbations**, including (a) similarities between original and perturbed text-features, and (b) distribution of original text-feature, perturbed text-features, and image-features.

**Perturbing ratio**. As an important factor, perturbing how many visual characters $\{v_j^p\}^m$ to simulate a spurious category seems to be very sensitive. A too-small ratio may lead to noneffective perturbation and cause invalid candidates from ID data, while too-large ratios can distort the spurious category description and bring invalid candidates from far OOD. In fact, as previous analysis in Fig. A3 implies perturbing one word/token $v_j^p$ can produce **effective** deviation on the encoded text feature $\mathring{\mathbf{w}}_k^p$, Tab. A1 suggests that one-word-perturbation is **effective enough** to perform perturbation guidance as described in Sec. 3.2. Perturbing 4 or more words even leads to performance degradation, which means severely perturbed contexts may choose the noisy OOD candidates (*e.g.*, random noise), making the learned spurious context unable to capture the true spurious OOD samples and further describe the category boundary.

Table A1: Ablation for perturbing ratio on ImageNet-100 benchmark.

| Perturbing ratio | FPR95↓ | AUROC↑ |
|:---:|:---:|:---:|
| 1 / 16 | 10.31 | **97.82** |
| 2 / 16 | **10.27** | 97.81 |
| 4 / 16 | 10.47 | 97.78 |
| 8 / 16 | 11.02 | 97.73 |
| 16 / 16 | 11.70 | 97.62 |

**Perturbing position**. Another hyper-parameter to perform perturbation on perceptual contexts is the perturbing position, *i.e.*, which word/token to perturb? In Sec. 3.2 from the manuscript, we randomly choose the word in perceptual contexts to perturb at each training iteration. Indeed, deciding the optimal word to perturb is challenging, especially when the learned word embeddings do not correspond to actual words in natural language. Though, we have made a simple step to find the "optimal" word/token in the embedding space. After computing the prototype vector by averaging the 16 learned words, we take the most distant (denoted as *MaxDist*) or closed (denoted as *MinDist*) word for perturbation guidance for spurious OOD syntheses. However, as shown in Tab. A2, the final OOD detection performance even gets worse. Thus, we argue that randomly choosing the words to perturb at each iteration is still an effective way. After several iterations, we can statistically perturb every word to guide the spurious sample generation, covering the optimal situation.

Table A2: Ablation for perturbing position on ImageNet-100 benchmark.

| Method | FPR95↓ | AUROC↑ |
|--------|--------|--------|
| MaxDist | 10.42 | 97.75 |
| MinDist | 10.86 | 97.73 |
| Random | **10.31** | **97.82** |

## A.3 Cross-ID-Domain Generalization

As indicated in the manuscript, the precise category boundary learned by our method shows robust OOD performance when the ID data is shifted. In fact, the shifted ID classification can be further boosted by our proposed integrated inference strategy (Eq. (5) in the manuscript), as shown in Tab. A3. It implies the regularization item $\gamma$ successfully modulates the relative similarities between input images and learned perceptual descriptions for each category, leading to more precise boundaries.

Table A3: Additionally improved ID accuracy on shifted datasets.

| Method | Target Datasets | | |
|--------|-----------|-----------|---------------|
| | ImageNet-A | ImageNet-R | ImageNet-Sketch |
| CATEX | 50.87 | 76.67 | 48.59 |
| +IntegInfer. | **50.98** | **76.72** | **48.65** |

However, our method only takes the secondary place on ImageNet-Sketch [56] on both ID classification (inferior to NPOS [50]) and OOD detection (inferior to MCM [38]). It is mainly because of the huge domain gap between vanilla ImageNet-1K [11] and shifted ImageNet-Sketch. As shown in Fig. A4, compared to the shifted ImageNet-A [20] and ImageNet-R [23], images from ImageNet-Sketch only preserve objects' shape and main texture, while the color information is totally vanished. We leave the generalization to heavily-shifted ID datasets as future work.

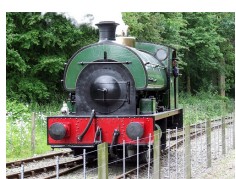 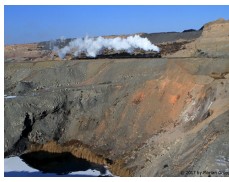 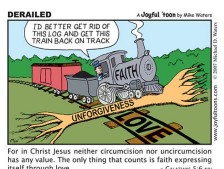 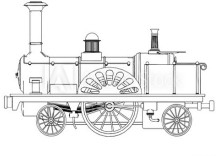

Figure A4: Left to right: examples from ImageNet, ImageNet-A, ImageNet-R, and ImageNet-Sketch.

## A.4 Cross-ID-Task Generalization

To verify the efficacy of our proposed framework, we conduct a category-extended experiment in Sec.4.1 and Tab.3. Here more implementation details are provided for reproducibility.

Given two models independently trained on the separated ImageNet-100 (I) and ImageNet-100 (II), how to test them on the union ImageNet-200 (I ∪ II) with our CATEX is simple. In the vision-language prompt-tuning framework, the image-encoder $\mathcal{I}$ and text-encoder $\mathcal{T}$ are frozen, and we only learn the perceptual and spurious contexts (*i.e.*, $\mathbf{v}^p$ and $\mathbf{v}^s$). And the $l_2$-normalized text-feature can be pre-extracted with the 100 category names in each subset, taking the perceptual descriptions for example, which are denoted as $\{\mathbf{w}_{\mathrm{I},k}^p = \mathcal{T}(\mathbf{v}_{\mathrm{I},k}^p; \mathrm{CLS}_{\mathrm{I},k})\}_{k=1}^{100}$ and $\{\mathbf{w}_{\mathrm{II},k}^p = \mathcal{T}(\mathbf{v}_{\mathrm{II},k}^p; \mathrm{CLS}_{\mathrm{II},k})\}_{k=1}^{100}$. During inference, one may concatenate the 200 text-features together as $\{\mathbf{w}_k^p\}_{k=1}^{200}$. Given an input image $I$, the $l_2$-normalized image-feature is extracted by $\mathbf{x} = \mathcal{I}(I)$, and the perceptual image-text similarities are computed as $\mathbf{s}^p = [\langle \mathbf{w}_1^p, \mathbf{x} \rangle, \langle \mathbf{w}_2^p, \mathbf{x} \rangle, \cdots, \langle \mathbf{w}_{200}^p, \mathbf{x} \rangle] \triangleq [s_1^p, s_2^p, \cdots, s_{200}^p]$. Similarly, the spurious similarities become $\mathbf{s}^s = [s_1^s, s_2^s, \cdots, s_{200}^s]$. Then we can leverage the measurement defined in Eq. (5) for both ID classification and OOD detection.

As for the competitors, (*e.g.*, VOS [15] and NPOS [50]), two image-encoders are trained separately (denoted as $\mathcal{I}_{\mathrm{I}}$ and $\mathcal{I}_{\mathrm{II}}$). And for each input image $I$, there are two corresponding image-features:

$\mathbf{x}_{\mathrm{I}} = \mathcal{I}_{\mathrm{I}}(I)$ and $\mathbf{x}_{\mathrm{II}} = \mathcal{I}_{\mathrm{II}}(I)$. Consequently, there also two sets of image-text similarity vector: $\mathbf{s}_{\mathrm{I}} = [\langle \mathbf{w}_1, \mathbf{x}_{\mathrm{I}} \rangle, \langle \mathbf{w}_2, \mathbf{x}_{\mathrm{I}} \rangle, \cdots, \langle \mathbf{w}_{200}, \mathbf{x}_{\mathrm{I}} \rangle] = \{\langle \mathbf{w}_k, \mathbf{x}_{\mathrm{I}} \rangle\}_{k=1}^{200}$ and $\mathbf{s}_{\mathrm{II}} = \{\langle \mathbf{w}_k, \mathbf{x}_{\mathrm{II}} \rangle\}_{k=1}^{200}$ (the superscript $p$ is hidden for simplicity). For compatibility, we choose the one for ID classification and OOD detection according to its highest image-text similarity. $\mathbf{s} = \begin{cases} \mathbf{s}_{\mathrm{I}} & max(\mathbf{s}_{\mathrm{I}}) > max(\mathbf{s}_{\mathrm{II}}) \\ \mathbf{s}_{\mathrm{II}} & \texttt{otherwise} \end{cases}$. Now, the performance of our method and other rivals are evaluated under the same measurements.

Note that since we only take one image encoder throughout, the inference time is fixed (because the text-features can be pre-extracted). In contrast, applying other methods brings multiple time cost (*e.g.*, twice slower than ours in this case). When the training subsets extend intensely (*e.g.*, from ImageNet-1K to ImageNet-21K in our manuscript), our method still keeps a fast speed (*e.g.*, 100FPS on V100) during inference, which can even enable real-time applications in practice.

### A.5 Software and Hardware

We use Python 3.7.13 and PyTorch 1.8.1, and 2 NVIDIA V100-32G GPUs.

## B Additional Experiments

### B.1 Comparison to CoCoOp

Compared with the vanilla vision-language prompt tuning method CoOp [71], the newer version CoCoOP [70] was explicitly designed to deal with out-of-distribution issues through image conditional contexts. However, Tab. A4 indicates that CoCoOp is surprisingly much worse than our method. The reason may be that during training, CoCoOp only takes ID images as context conditions, while neither OOD samples nor OOD contexts are involved. Thus, when employed in the open-world and asked to reject OOD samples not belonging to any ID category, CoCoOp failed. The image conditions even aggravate the overconfidence in OOD samples.

Table A4: Comparison to CoCoOp on ImageNet-100 benchmark.

| Method | ID-ACC↑ | FPR95↓ | AUROC↑ |
|--------|---------|--------|--------|
| CoCoOp | 92.90 | 39.22 | 92.69 |
| **Ours** | **94.12** | **10.31** | **97.82** |

### B.2 Combination with Post-hoc Enhancements

Recently, post-hoc OOD detection methods that enhance the single-vision-modal networks (*e.g.*, ResNet [19] and ViT [14]) have been widely studied [9, 46, 47, 73, 12]. In this section, we make a step towards combining vision-language models with previous post-hoc enhancements for better OOD performance. The results are shown in Tab. A5, where ReAct [46] achieves a remarkable improvement. It indicates that pruning the extreme feature values according to the unified distributional statistics may be more suitable for VLMs to reduce the overconfidence on OOD samples. We hope this can bring new insights to the community.

Table A5: Combination with post-hoc methods.

| Cmobine | ImageNet-100 | | ImageNet-1K | |
|---------|--------------|--------|-------------|--------|
| | FPR95↓ | AUROC↑ | FPR95↓ | AUROC↑ |
| None | 10.31 | 97.82 | 29.66 | 93.48 |
| ReAct [46] | **10.06** | 97.82 | **27.56** | **93.77** |
| BATS [73] | 10.16 | **97.84** | 29.37 | 93.59 |
| ASH [12] | 10.19 | 97.81 | 29.14 | 93.27 |

## B.3 Inference-time perturbation

Now that every perturbation can directly produce the description (*i.e.*, text-feature) of an unknown spurious category, one may try to take the perturbed description as a substitute for the learned spurious contexts to execute OOD detection. That is, use the perturbed $\mathring{\mathbf{w}}_k^p$ to replace the learned $\mathbf{w}_k^s$ in Eq. (5) in the manuscript. And the results are shown in Tab. A6, where ImageNet-100 [38] is the ID dataset. Given a baseline model [71] learned with perceptual contexts only, simply using the perturbed descriptions (denoted as +*Perturb-Desc.*) brings slight improvements (*e.g.*, 0.2% decrease on FPR95). The insignificant advantage is not due to the limited capacity of only one perturbed description for each ID category. Because ensembling [38] several perturbed descriptions for an ID category at once (denoted as +*Perturb-Ensem.*) dose not bring remarkable improvements. In contrast, our proposed CATEX can significantly enhance the OOD detection performance, which demonstrates it is still necessary to explicitly learn the spurious contexts for each ID category.

Table A6: Comparison with directly using perturbed descriptions for OOD detection.

| Method | FPR95↓ | AUROC↑ |
|---|---|---|
| baseline [71] | 13.07 | 97.42 |
| +Perturb-Desc. | 12.84 | 97.43 |
| +Perturb-Ensem. | 12.87 | 97.45 |
| **CATEX** (Ours) | **10.31** | **97.82** |

## B.4 Experiments on CIFAR Benchmarks

To further verify the robustness of our method, we conduct additional experiments on CIFAR-10 and CIFAR-100 datasets, and evaluate the OOD detection performance on SCOOD [66] benchmark. We train our CATEX for 20 epochs, and the other settings are the same as Sec.4 in the manuscript. The results are shown in Tab. A7 and Tab. A8, where "Surr." means the extra TinyImages80M [51] is adopted for surrogate OOD training set. Accordingly, our CATEX consistently ourperforms the competitors as well, and even surpasses those who adopts the extra OOD training data. It implies the pre-trained knowledge for large-scale CLIP [41] model leveraged by our method is capable enough to detect the OOD samples in the open-world. The efficacy of our CATEX is further demonstrated.

Table A7: Performance on CIFAR-10.

| Method | Surr. | ID-ACC ↑ | FPR95↓ | AUROC↑ |
|---|---|---|---|---|
| MCM [38] | ✗ | 90.79 | 23.14 | 94.68 |
| ODIN [32] | ✗ | 95.36 | 52.00 | 82.00 |
| Energy [33] | ✗ | 95.36 | 50.03 | 83.83 |
| OE [22] | ✔ | 94.90 | 50.53 | 88.93 |
| UDG [66] | ✔ | 94.71 | 36.22 | 93.78 |
| **CATEX** | ✗ | **95.57** | **21.17** | **95.33** |

Table A8: Performance on CIFAR-100.

| Method | Surr. | ID-ACC ↑ | FPR95↓ | AUROC↑ |
|---|---|---|---|---|
| MCM [38] | ✗ | 66.91 | 71.93 | 79.39 |
| ODIN [32] | ✗ | 81.84 | 81.89 | 77.98 |
| Energy [33] | ✗ | 81.84 | 83.66 | 79.31 |
| OE [22] | ✔ | 81.31 | 80.06 | 78.46 |
| UDG [66] | ✔ | 80.89 | 75.45 | 79.63 |
| **CATEX** | ✗ | **81.99** | **67.95** | **84.04** |

## B.5 Error Bars

To verify the robustness, we repeat the training of our method and the rivals on ImageNet-100 [38] with CLIP-B/16 for 3 times, and the results are shown in Tab. A9. Our CATEX consistently outperforms the rivals on OOD detection by a significant margin.

Table A9: Error Bars on ImageNet-100 after 3 runs

| Method | ACC↑ | FPR95↓ | AUROC↑ |
|---|---|---|---|
| MSP [18] | **94.77** (±0.05) | 41.90 (±0.61) | 93.38 (±0.05) |
| Energy [33] | **94.77** (±0.05) | 31.89 (±0.50) | 94.53 (±0.18) |
| VOS [15] | 94.75 (±0.07) | 24.48 (±0.71) | 96.04 (±0.36) |
| NPOS [50] | 94.34 (±0.12) | 17.32 (±0.87) | 96.46 (±0.13) |
| **CATEX** | 94.11 (±0.03) | **10.97** (±0.79) | **97.75** (±0.07) |

## C  Additional Analysis

**Performance improvement.** To further evaluate the improvement brought by our method (*e.g.*, 8% decrease of FPR95 against NPOS), we conduct a comparative experiment on ImageNet-1K. To provide a unified analysis across two models, we take a third-party ResNet-50 model [64] (pre-trained on ImageNet-1K classification only) to produce the Maximum SoftMax Probability for each OOD sample that is correctly detected by our CATEX while wrongly viewed as ID samples by NPOS. According to Fig. A5, our method consistently improves the OOD detection on each interval, where the high-probability OOD (generally hard samples) detection is significantly enhanced. It indicates that properly leveraging the prior knowledge from pre-trained VLMs can alleviate the OOD problem when the fine-tuned visual features are indistinguishable, which is consistent with our motivation.

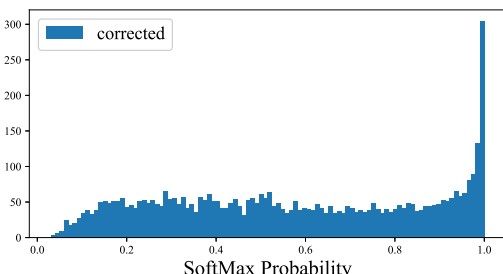

Figure A5: Corrected OOD detections compaired with NPOS. The softmax probability predictions on those OOD samples are produced by another pre-trained ResNet-50 [64] classifier.

**Failure cases.** As our method still gets 29% FPR95 on ImageNet-1K, we provide some failure cases in Fig. A6, which can be summarized into three kinds:

- Noisy label, where the ID objects (*e.g.*, *dam*) also exists in some OOD images from the test set. And the dataset composition may need a further examination.
- Similar texture, shared by some OOD samples (*e.g.*, *flower*) against ID images (*e.g.*, *starfish*), and the pre-trained encoders of CLIP are unable to distinguish their features. Applying image-level spurious OOD syntheses (*e.g.*, image attacks [35, 72]) may reduce the texture-bias.
- Same background (*e.g.*, *sky*) that seizes a large proportion of the image may lead to similar feature representations. Adopting image-level automatic masking techniques [2, 68] to synthesize spurious OOD samples may alleviate such problem.

Similar failure cases are also observed in recent SOTA methods, which reveal the unsolved challenges of OOD detection and suggest the potential directions for future works.

## D  Datasets and Baselines

For reproducibility, we present the details of datasets and baselines as follows.

**ImageNet-100 (I).** Following MCM [38], we take the randomly-sampled 100 classes from ImageNet-1K [11] as the ImageNet-100 (I) subset, which contains the following categories: n03877845, n03000684, n03110669, n03710721, n02825657, n02113186, n01817953, n04239074, n02002556, n04356056, n03187595, n03355925, n03125729, n02058221, n01580077, n03016953, n02843684, n04371430, n01944390, n03887697, n04037443, n02493793, n01518878, n03840681, n04179913, n01871265, n03866082, n03180011, n01910747, n03388549, n03908714, n01855032, n02134084, n03400231, n04483307, n03721384, n02033041, n01775062, n02808304, n13052670, n01601694, n04136333, n03272562, n03895866, n03995372, n06785654, n02111889, n03447721, n03666591, n04376876, n03929855, n02128757, n02326432, n07614500, n01695060, n02484975, n02105412, n04090263, n03127925, n04550184, n04606251, n02488702, n03404251, n03633091, n02091635, n03457902, n02233338, n02483362, n04461696, n02871525, n01689811, n01498041, n02107312, n01632458, n03394916, n04147183, n04418357, n03218198, n01917289, n02102318, n02088364, n09835506, n02095570, n03982430, n04041544, n04562935, n03933933, n01843065, n02128925, n02480495, n03425413, n03935335, n02971356, n02124075, n07714571, n03133878, n02097130, n02113799, n09399592, n03594945.

**ImageNet-100 (II).** Disjoint from ImageNet-100 (I), ImageNet-100 (II) contains another 100 classes randomly sampled from ImageNet-1K: n02096177, n03769881, n01629819, n04033995, n04357314, n02101388, n02328150,

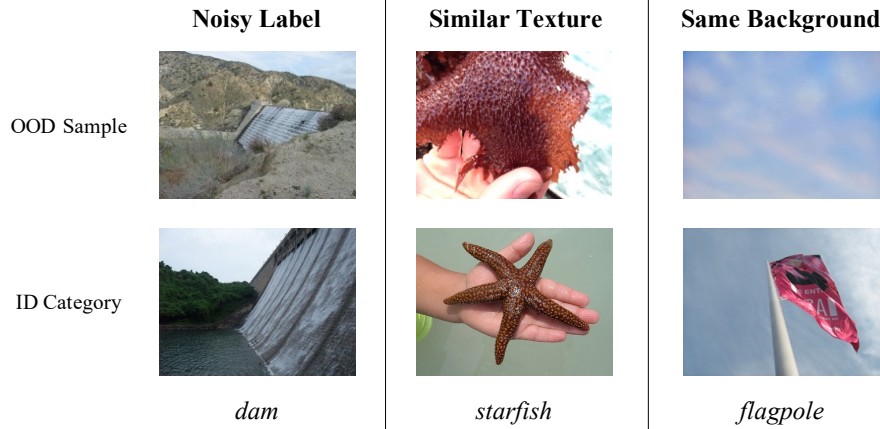

Figure A6: Failed OOD detections of our CATEX

n03729826, n02655020, n01985128, n02109525, n07715103, n02099429, n04517823, n02088632, n03207743, n03657121, n02948072, n02106662, n01631663, n09229709, n03793489, n03776460, n07860988, n02129604, n03483316, n02107574, n07716358, n04208210, n02107908, n04372370, n02119022, n12144580, n01693334, n04548280, n03785016, n03535780, n03599486, n02859443, n04335435, n02110341, n03902125, n04146614, n01774750, n03314780, n03045698, n01697457, n02869837, n02276258, n04081281, n03956157, n02487347, n04311174, n02094114, n04409515, n03028079, n03384352, n04532106, n02087394, n04612504, n02100583, n11939491, n02107142, n01669191, n12998815, n04522168, n02894605, n03529860, n10148035, n01677366, n03775071, n03208938, n04238763, n02363005, n02804414, n02106382, n03950228, n02128385, n02028035, n04099969, n02481823, n01729322, n02939185, n02483708, n04162706, n03857828, n02093647, n02927161, n03160309, n02840245, n03920288, n07871810, n04404412, n03947888, n04509417, n02086910, n02256656, n02412080, n02410509, n03584829.

**ImageNet-21K.** The ImageNet-21K dataset on which we conduct the category-extended experiment is the official winter 2021 released version [*]. For pre-processing, we follow Ridnik *et al* [42] to clean invalid classes, allocating 50 images per class for validation, and crop-resizing all the images to 224 resolution. Training settings are the same as Sec.4 in our manuscript.

**OOD datasets.** Following the literature [59, 50, 60, 38], we mainly consider subsets of `iNaturalist` [53], `SUN` [65], `Places` [69], and `Texture` [8] as the OOD datasets, which contains 35640 images in total.

**Baselines.** To evaluate the baselines on our experiment settings, we re-implement the most representative and relevant methods, including MSP [21, 18], Energy [33], VOS [15], and NPOS [50]. For a fair comparison, we train all the baselines with NPOS's codebase [†], and only fine-tune the last two transformer blocks of image encoder [50].

- For MSP and Energy, we train a single model with standard cross-entropy loss function for ID classification only, and infer with respective OOD metrics.
- For VOS, we take the likelihood-based sampling strategy to generate spurious OOD syntheses, and train the model with uncertainty regularization as suggested [15].
- For NPOS, we take the non-parametric distance-based sampling strategy to generate spurious OOD syntheses, and train the model with open-set ERM as suggested [50].

---

[*]https://image-net.org/
[†]https://github.com/deeplearning-wisc/npos

