# OpenReview forum: "Category-Extensible Out-of-Distribution Detection via Hierarchical Context Descriptions"
_NeurIPS.cc/2023/Conference — NeurIPS 2023 poster_

### Official Review · Reviewer_194m · 2023-06-19

**Soundness:** 2 fair
**Presentation:** 3 good
**Contribution:** 3 good
**Rating:** 6
**Confidence:** 3

**Summary:**

The paper presents a new technique to improve OOD, as well as IID, prediction for a pre-trained Language Vision Model (LVM). The heart of the proposed method is to learn both an ID context (i.e. perceptual context) and an OOD context (i.e. spurious context) to improve the classification of both ID classes and OOD samples. The paper proposes a new loss function to combine the ID and OOD losses together and a sampling strategy to produce spurious samples.

**Strengths:**

The strengths of the paper are in its empirical validation and the originality around combining insights from different works. The proposed technique in the paper shows strong empirical performance, including ablation testing, on the main OOD tasks and measures. The quality of this validation well supports the claims made in the article about the need to consider both ID and some kind of spurious context when doing open-world detection.

The paper also combines elements from previous work, like the idea of having perturbed examples from VOS and NPOS, and the idea of learning prompts in the text vector space from Learning to Prompt into one framework.


**Weaknesses:**

Despite its strong empirical validation, the paper does have some weaknesses in its clarity and novelty. Beginning with novelty, the proposed technique of CATEX seems to be only an incremental improvement on NPOS (e.g., creating perturbed examples as part of training for OOD) and directly uses the technique from Learning to Prompt with only a change in the loss function. In essence, the paper doesn’t present any insight that the NPOS/VOS papers already presented, namely the inclusion of perturbed samples into learning for an LVM can help with OOD performance. If the paper were more explicit, especially in the methods section and discussion section, about how the proposed method differs from previous ones, it would help for establishing the novelty. For example, I believe both VOS and NPOS train the underlying CLIP model, while the proposed technique of CATEX uses the Learning to Prompt technique of training a lightweight layer on top of the CLIP model. Such a change seems to balance between being good at ID tasks, while not distorting the feature space.

In terms of clarity, there is not enough detail in the methods section to both deal with the aforementioned novelty issues as well as to fully understand the training process and the perturbation guidance. For the perturbation guidance, lines 174-176 make it sound like its changing the actual words or tokens (as is done in the Kwon et al. article with masked language modeling), versus the embeddings of the text, as is done in Learning to Prompt article. If the proposed technique is actually masking the tokens, versus changing the context vectors in the embed space, then how is the training done to optimize the text, as the method in Learning to Prompt only deals with a vector space? Also, does perturbation by masking tokens fully make sense? For example, is the perturbation of “a photo of a dog” to “a [MASK] of a dog” or “a photo of a [MASK]” really a meaningful perturbation for the spurious context? I wish there was something like a walkthrough example of the perturbation as well as some more explanation of the training method and intuition behind the perturbation to better understand the contribution of the work.

Finally, there are a couple of areas where the writing could be improved. There are some sentences throughout that need to be proofread for grammar. For example, the last sentence of the abstract is a run-on sentence and the sentence on lines 48-50 is unclear in what its trying to say.

-------- Following author's responses --------------

I believe the authors have significantly addressed my concerns about the perturbation guidance. They have both added additional explanations as well as done some additional experimentation around ideas like how many tokens to mask.


**Questions:**

1. How does the proposed method find optimal perturbations of the textual input? How does the method decide on which tokens or words to mask?

2. How could the proposed method be used when you have no labeled data (i.e. a true zero-shot setting) to improve performance? Or can it? For example, can perturbations be included in at inference time – combined with the OOD scoring function – to do zero-shot labeling?

3. What is the performance of the proposed method versus CoCoOP? While the paper does investigate the performance of its proposed method versus CoOp, and rightly concludes its does better with OOD, it does not evaluate the performance of its proposed method against the newer version of the CoOP method (i.e. CoCoOP) which was explicitly designed to deal with OOD issues of CoOP.


-------- Following author's responses --------------
I believe the authors have answered all three of my questions. In particular, I found their answer around 2, of how to use the proposed method to improve the zero-shot performance of LVMS, to be very interesting and notable. I also find it quite interesting that there was such a performance gap between the proposed method and CoCoOP.

**Limitations:**

The authors have addressed nearly all of their limitations and addressed those dealing with societal impact. The only limitation they have not addressed is that the proposed method still requires labeled (or captioned) data in order to work and cannot work in a zero-shot setting as was the promise of CLIP. I welcome the author’s reply on this, as I am not sure if the proposed method couldn’t be used without labeled data.

---

> ### Author Rebuttal · Authors · 2023-08-09
>
> We are pleased that the reviewer recognizes our work‘s originality and empirical validity.
>
> We appreciate your experienced comments and valuable suggestions, which are addressed below in detail:
>
> > Q1: No insights beyond NPOS/VOS. Clarifying the method difference helps establish novelty.
>
> Thanks for your kind comment. As follows, we provide at least three new insights.
> 1. **Large-scale pretrained VLM itself provides significant advances in OOD detection**. Huge corpora bring powerful feature representation capability, which is the key to detecting diverse unknown OOD samples beyond the limited ID datasets. Furthermore, massive paired image-text data builds comprehensive multi-modal prior knowledge, offering valuable extra information for OOD detection.
> 2. **Previous methods have not fully utilized VLM's advances**. NPOS/VOS optimize a more compact ID feature space via **random** OOD syntheses, but the generalized feature space is distorted and the image-text prior knowledge is not utilized. Besides, vanilla CoOp lacks explicit descriptions for OOD samples, which hinders further OOD detection.
> 2. **Our method leverages VLM's advances via hierarchical contexts**. Freezing CLIP's encoders to maintain the generalized feature space, we learn two contexts (i.e., perceptual and spurious) to hierarchically describe each ID category's boundary. Moreover, to exploit VLM's prior knowledge, a novel textual-perturbation-guided approach is developed to generate OOD samples to learn the contexts.
>
> Hence, this paper is specifically designed for OOD detection with VLMs, but not an incremental improvement on NPOS.
>
> > Q2: About training process and perturbation guidance. How to optimize the text? Does the perturbation of “a [MASK] of a dog” or “a photo of a [MASK]” makes sense?
>
> Thanks for your comment. First, in the training procedure, the perturbation itself only serves to generate OOD samples and no gradient is involved. Detailed diagrams (**Figure A1-A2**) are provided in the uploaded PDF file to illustrate the whole training procedure.
>
> Second, rather than the initial prompt template as "a photo of a", the perturbation is applied to the tuned contexts [V1][V2]...[Vm]+[CLS], where [CLS] is the fixed class name (e.g., dog or goldfish). As each word [Vi] has a certain semantic meaning $^{[1]}$, let's view the learned contexts as (even though not) "yellow fish long flowing tail" for [CLS] as [*goldfish*]. Thus, the perturbations of "[MASK] fish long flowing tail" and "yellow fish long flowing [MASK]" both make sense, which eliminate some kinds of visual features of [*goldfish*], and generally describe another spurious OOD category against the certain ID category.
>
> [1] Kaiyang Zhou. Learning to Prompt for Vision-Language Models, IJCV 2022.
>
> > Q3: The writing can be improved. Lines 48-50 are unclear.
>
> Thanks to this, we have made a throughout proofreading to improve readability.
>
> In particular, the core idea conveyed by lines 48-50 is that even though fine-tuning CLIP's encoder may boost the performance (e.g., classification accuracy on ImageNet), CLIP's generalized feature space is destroyed. So test data shifts (e.g., ImageNet-R) will cause severe performance degradation.
>
> > Q4: How to find optimal perturbations of the textual input?
>
> A thoughtful comment! Indeed, deciding the optimal word to perturb is challenging, especially when the learned word embeddings do not correspond to actual words in natural language.
> Though, we have made a simple step to find the "optimal" word/token in the embedding space. After computing the prototype vector by averaging the 16 learned words, we take the most distant (denoted as *MaxDist*) or closed (denoted as *MinDist*) word for perturbation guidance. However, as shown below, the results even get worse.
>
> |Method|FPR95↓|AUROC↑|
> |:----:|:----:|:----:|
> |MaxDist|10.42|97.75|
> |MinDist|10.86|97.73|
> |Random|**10.31**|**97.82**|
>
> So, we argue that randomly choosing the words to perturb at each iteration is still an effective way. After several iterations, we can statistically perturb every word to guide the spurious sample generation, covering the optimal situation.
>
> > Q5: How to deploy without labeled data? Can perturbations be used to do zero-shot labeling?
>
> An insightful suggestion! Our method actually CAN be used for zero-shot labeling. For ImageNet-1K, since CLIP's default prompt "a photo of a [CLS]" contains no visual info, we adopt the visual description from LLM $^{[2]}$ as the initial prompt input (denoted as VisDesc). Then, we randomly mask the visual description to simulate spurious contexts, and adopt Eq.(5) in our manuscript to regularize image-text similarities for classification. The results shown below indicate our method boosts zero-shot classification without any training cost.
>
> |Method|Prompt|ACC-Top1|ACC-Top5|
> |:----:|:------:|:-----:|:----:|
> |CLIP|A photo of a goldfish.|63.50|88.99|
> |VisDesc|A yellow fish with a long flowing tail is goldfish.|65.47|90.14|
> |Ours|(plus) A [MASK] fish with a long [MASK] [MASK] is goldfish.|**65.83**|**90.36**|
>
> It provides another new insight that in the category-extensible setting or a true zero-shot classification scenario, explicitly constructing spurious contexts can perform the one-class OOD detection task, as other categories can also be viewed as OOD for a certain ID category.
>
> [2] Sachit Menon, Visual Classification via Description from Large Language Models. ICLR 2023.
>
> > Q6: Comparison to CoCoOP?
>
> On ImageNet-100, CoCoOp is surprisingly much worse than our method. The reason may be that during training, CoCoOp only takes ID images as context conditions, while neither OOD samples nor OOD contexts are involved. Thus, when employed in the open-world and asked to reject OOD samples not belonging to any ID category, CoCoOp failed. The image conditions even aggravate the overconfidence in OOD samples.
>
> |Method|ACC↑|FPR95↓|AUROC↑|
> |:----:|:--:|:---:|:----:|
> |CoCoOp|92.90|39.22|92.69|
> |Ours|**94.12**|**10.31**|**97.82**|

---

> > ### Comment · Reviewer_194m · 2023-08-13
> > **Reponse to Author's Rebuttal**
> >
> > I really appreciate the authors' addressing of my questions and attempting to better elucidate how the perturbation is done. In particular, I am rather impressed by the performance of the proposed method against CoCoOP (which is supposed to handle OOD better) and how the method can be used to improve zero-shot image classification. This later answer, in particular, raised my estimation of this paper rather significantly. I do still wish there was some kind of easier-to-follow flow chart or even pseudo-code on how to implement the perturbation of CATEX for use in training and zero-shot classification, though.
> >
> > Given the authors' responses, I have raised my rating and do believe the paper should be given serious consideration for acceptance.

---

> > > ### Author Response · Authors · 2023-08-14
> > >
> > > We thank the reviewer again for evaluating our work and carefully reading our response. Your constructive comments and insightful suggestions do make our work much stronger, and we are really encouraged that you can boost our paper.
> > >
> > > On the other hand, we are sorry that in the discussion stage, it is not allowed to provide new charts or figures to illustrate our method. Therefore, to answer your following-up questions on how our perturbation is implemented in (1) training our CATEX and (2) performing zero-shot classification, we provide the pseudo-codes below:
> > >
> > > > 0. How is the perturbation implemented?
> > >
> > > The perturbation itself is simple. For a context $\mathbf{v}$, the perturbation can be expressed as:
> > >
> > > 1. Take the $m$ (e.g., 16) learned/pre-defined tokens/words $\mathbf{v} = [v_1;v_2;\cdots;v_m]$
> > > 2. Generate masking/noise perturbations $u$
> > > 3. Randomly perturb one (or more) token/word $\mathring{\mathbf{v}} = [v_1;u;\cdots;v_m]$
> > >
> > > The above process is formulated as $\mathring{\mathbf{v}} = \mathcal{P}(\mathbf{v})$. With the text-encoder $\mathcal{T}$ and class name $[\texttt{CLS}]$, the perturbed text embedding is encoded as $\mathring{\mathbf{w}} = \mathcal{T}(\mathring{\mathbf{v}}, [\texttt{CLS}])$.
> > >
> > > > 1. How is the perturbation implemented in the vanilla training procedure?
> > >
> > > Our method trains a pair of perceptual context $\mathbf{v}^p$ and spurious context $\mathbf{v}^s$ for each category, with real in-distribution samples $\lbrace\mathbf{x_i}\rbrace$ and generated OOD samples (guided by the perturbation) $\lbrace\tilde{\mathbf{z}}_j\rbrace$.
> > > The training procedure can be expressed as:
> > >
> > > 1. Perturb the perceptual context $\mathring{\mathbf{v}}^p = \mathcal{P}(\mathbf{v}^p)$
> > > 2. Generate random OOD samples $\lbrace\tilde{\mathbf{z}}_j^\prime\rbrace = \mathcal{G}(\lbrace\mathbf{x_i}\rbrace)$
> > > 3. Use perturbation to select OOD samples $\lbrace\tilde{\mathbf{z}}_j\rbrace = \mathcal{F}(\lbrace\tilde{\mathbf{z}}_j^\prime\rbrace, \mathring{\mathbf{v}}^p)$ via Eq.(4)
> > > 4. Encode text embeddings $\mathbf{w}^p = \mathcal{T}(\mathbf{v}^p)$, $\mathbf{w}^s = \mathcal{T}(\mathbf{v}^s)$
> > > 5. Train with loss functions $\mathcal{L}(\mathbf{w}^p, \mathbf{w}^s, \lbrace\mathbf{x_i}\rbrace, \lbrace\tilde{\mathbf{z}}_j\rbrace)$ as Eq.(1) and Eq.(3)
> > >
> > > The random OOD sample generator $\mathcal{G}$ can be distance-based $^{[1]}$, density-based $^{[2]}$, etc.
> > > Kindly note that the perturbation guidance itself does not involve gradient back-propagation, and we only optimize perceptual contexts $\mathbf{v}^p$ and spurious contexts $\mathbf{v}^s$.
> > >
> > > > 2. How is the perturbation implemented to help zero-shot classification?
> > >
> > > For zero-shot classification, we apply perturbation on pre-defined category descriptions (viewed as perceptual context $\mathbf{v}^p$) to simulate the spurious context $\hat{\mathbf{v}}^s$. Give an input image $x$, the classification process can be expressed as:
> > >
> > > 1. Perturb the perceptual context $\hat{\mathbf{v}}^s = \mathcal{P}(\mathbf{v}^p)$
> > > 2. Encode text embeddings $\mathbf{w}^p = \mathcal{T}(\mathbf{v}^p)$, $\hat{\mathbf{w}}^s = \mathcal{T}(\hat{\mathbf{v}}^s)$
> > > 3. Get initial image-text similarity $s = \langle \mathbf{w}^p, x \rangle$
> > > 4. Compute regularization item $\gamma = \mathcal{R}(\mathbf{w}^p, \hat{\mathbf{w}}^s, x)$ via Eq.(5)
> > > 5. Compute regularized similarity $r = s \times \gamma$
> > > 6. Determine the category $k = argmax_{k}  \lbrace r_k \rbrace $
> > >
> > > We will add more detailed pseudo-codes or flow charts in the revised paper.
> > >
> > > For reproducibility, the source code will be released upon acceptance. And we are also happy to answer any remaining or follow-up questions to clarify our method.
> > >
> > >
> > > [1] Leitian Tao. Non-Parametric Outlier Synthesis, ICLR 2023.
> > >
> > > [2] Xuefeng Du. VOS: Learning What You Don't Know by Virtual Outlier Synthesis, ICLR 2022.
> > >
> > > ---
> > >
> > > Best regards,
> > >
> > > Authors

---

### Official Review · Reviewer_R97w · 2023-07-04

**Soundness:** 2 fair
**Presentation:** 2 fair
**Contribution:** 2 fair
**Rating:** 6
**Confidence:** 3

**Summary:**

The paper proposes a method to incorporate perceptual context and spurious context to handle the OOD detection problem. The experimental results seem quite promising.

**Strengths:**

1. The results of the proposed method is obviously better than previous  OOD detection methods;
2.  The proposed method is quite interesting, which i think can be extended to other areas involving large-scale vision-and-language models.

**Weaknesses:**

1. The presentation of this paper is not very clear. For example, the authors mentioned "label bias" in the contributions part, but there is no explicit explanation about this so-called label bias, which I believe needs more clarifications;
2. The motivations for the proposed hierarchical context mechanism is not very clear. Why a single context cannot produce a precise classification boundary? Moreover, the spurious samples w.r.t. a specific ID category can have large variance, so using a single spurious context can encode such large intra-class variance?
3. It would be better if the authors can provide some visualizations of  the generated samples using the perturbation guidance.

**Questions:**

Kindly refer to  the weakness part

**Limitations:**

The authors can test that finetuning the encoders from the CLIP can produce what extent of the performance improvement for OOD detection.

---

> ### Author Rebuttal · Authors · 2023-08-09
>
> We are very cheerful that the reviewer finds our work interesting and transferable to other areas involving large-scale VLMs. We hope the detailed responses below can address your concerns.
>
> > Q1: The presentation of this paper is not very clear. For example, the mentioned "label bias" in the contributions part needs more clarification.
>
> We are sorry for that. The term "label bias" basically means the overconfidence in predicting unknown samples into known categories $^{[1][2]}$. To avoid ambiguity and reduce the burden of understanding, we will change the term "mitigate label bias" to the straightaway "detect OOD samples" in our contribution claim.
>
> In addition, we will make a throughout proofreading to improve readability and rigorousness.
>
> [1] Anh Nguyen. Deep neural networks are easily fooled: High confidence predictions for unrecognizable images. CVPR 2015.
>
> [2] Boxi Cao. Knowledgeable or educated guess? revisiting language models as knowledge bases. ACL 2021.
>
> > Q2: The motivation for the proposed hierarchical context mechanism is not very clear. Why a single context cannot produce a precise classification boundary? Moreover, the spurious samples w.r.t. a specific ID category can have large variance, so using a single spurious context can encode such large intra-class variance?
>
> Actually, a single perceptual context CAN produce a precise classification boundary in the closed-set, but CANNOT produce the precise category boundary in the open-set. As a result, the spurious OOD samples (similar to a certain in-distribution category) will lead to overconfidence predictions by the single perceptual context. Therefore, we propose to learn the spurious context to help perceptual context establish the precise category boundary.
>
> Besides, we recognize that for each ID category (e.g., cat), the spurious OOD samples are diverse (e.g., panthers, lions, etc). Thus, we aim to only describe the spurious sample surrounding the ID category, rather than the whole OOD space (which is too complicated). According to common sense, it is intuitive to use more spurious contexts ($w_k^p$) to describe better category boundaries. To verify it, we have tested the number of spurious contexts for each ID category (taking ImageNet-100 as the ID dataset), and the results are shown below:
>
> | Spurious Context Number | FPR95↓ | AUROC↑ |
> |:-----------------------:|:------:|:------:|
> |            1            |  10.31 |  97.82 |
> |            2            |  **10.21** |  97.86 |
> |            4            |  10.27 |  **97.88** |
> |            8            |  10.25 |  97.86 |
>
> It implies using more spurious contexts only leads to a 0.1% gain in performance. The reason may be that the learned 2 or more spurious contexts are too redundant without any constraints. To alleviate this problem, we simply add an orthogonal constraint (making the similarities between every two spurious contexts close to zero, denoted as *+orth*), and the OOD detection performance is significantly boosted:
>
>
> | Spurious Context Number | FPR95↓ | AUROC↑ |
> |:-----------------------:|:------:|:------:|
> |            1            |  10.31 |  97.82 |
> |         2 + orth        |  10.17 |  97.86 |
> |         4 + orth        |   9.89 |  **97.89** |
> |         8 + orth        |   **9.76** |  97.84 |
>
> Therefore, how to effectively and efficiently leverage more spurious contexts to better describe the category boundary deserves further exploration, and we view it as our future work.
>
> We will update the experiments and discussions.
>
> > Q3: It would be better if the authors can provide some visualizations of the generated samples using the perturbation guidance.
>
> Thanks for such a constructive suggestion. We have provided the visualizations in **Figure A1** in the uploaded PDF file. We hope it can better illustrate our approach.
>
> > Q4: The authors can test that finetuning the encoders from the CLIP can produce what extent of performance improvement for OOD detection.
>
> We have tested finetuning CLIP's encoders in the same category-extensible setting as Table 3 in our manuscript (separately training on two ImageNet-100 subsets (IN100-I, IN100-II), while directly testing on the merged ImageNet-200 (IN200)). Following NPOS $^{[3]}$, we only train the encoder's last two blocks, and the results are shown below.
>
> |          | IN100-I |        |        | IN100-II |        |        | IN200 |        |        |
> |:--------:|:-------:|:------:|:------:|:--------:|:------:|:------:|:-----:|:------:|:------:|
> | FineTune |   ACC↑  | FPR95↓ | AUROC↑ |   ACC↑   | FPR95↓ | AUROC↑ |  ACC↑ | FPR95↓ | AUROC↑ |
> |    No    |  94.12  |  10.31 |  97.82 |   94.42  |  7.91  |  98.31 | 89.61 |  13.13 |  97.19 |
> |    Yes   |  95.24  |  10.55 |  97.77 |   95.16  |  9.56  |  97.98 | 89.29 |  14.71 |  96.89 |
>
>
> Specifically, encoder-finetuning can indeed improve the in-distribution classification accuracy by 1% in each subset, but the OOD detection performance (i.e., FPR95 and AUROC) decreases. Moreover, when testing on the merged union set, the model obtains a lower ID classification accuracy with worse OOD detection performance. It is consistent with our motivation that freezing the encoders of large-scale vision-language models is necessary to describe the precision category boundaries.
>
> [3] Leitian Tao. Non-Parametric Outlier Synthesis, ICLR 2023.

---

> > ### Comment · Reviewer_R97w · 2023-08-11
> >
> > The author's response fully addresses my concerns and the added experiments further demonstrate the value of this work. So I decided to raise my rating to weak accept.

---

> > > ### Author Response · Authors · 2023-08-11
> > > **Thanks for your immediate feedback**
> > >
> > > We would like to express our sincere graititude to your immediate feedback. And we really appreciate that you can boost our paper. Your valuable comments do make our work much stronger.
> > >
> > > Best regards,
> > > Authors

---

### Official Review · Reviewer_oU43 · 2023-07-06

**Soundness:** 3 good
**Presentation:** 4 excellent
**Contribution:** 3 good
**Rating:** 6
**Confidence:** 5

**Summary:**

This paper contributes a new method for OOD detection by learning the precise category boundaries, specifically, the category boundaries are defined by one perceptual context and one spurious context, these two contexts are learned text embeddings for a frozen CLIP model, the spurious context is learned by perturbing the perceptual context.
Experiments on large scale OOD detection benchmarks show the effectiveness of the proposed method.


**Strengths:**

1. The experiments are comprehensive.
2. Because the proposed method does not change the parameter the frozen CLIP model, it is shown that the proposed method is more generalizable under multiple category extended scenarios.
3. The writing of the paper is clear.


**Weaknesses:**

1. The spurious context is learned by perturbing the perceptual context, and the perturbation is done by changing one word embedding, this could also be changing multiple word embeddings, but there are no ablations for this.
2. There is one paper[R1] explore similar idea of spuriour context (termed reciprocal points in [R1]), which I think should be cited and discussed.

[R1] Learning Open Set Network with Discriminative Reciprocal Points, ECCV 2020.

**Questions:**

1. I don't quite understand why the perceptual context is called hierarchical.


**Limitations:**

The limitations have been discussed in the paper.

---

> ### Author Rebuttal · Authors · 2023-08-09
>
> We are really encouraged that the reviewer thinks our method makes sense and the experiments are comprehensive.
>
> We appreciate your spot-on summary and constructive comments and suggestions, which we address below:
>
> > Q1: When learning spurious contexts, the perturbation could also be changing multiple word embeddings.
>
> Thanks for the suggestion. In fact, in the supplementary materials we have demonstrated that perturbing (masking is one of the perturbing approaches) one word/token in the perceptual context is **effective** to guide the OOD syntheses in section 3.2. Moreover, we have conducted a series of ablations to study how the masking ratio influences the final OOD detection performance. Taking ImageNet-100 as the ID dataset, we learn 16 words/tokens for each category, and randomly perturb 1/16 to 16/16 words (the *classname* is always preserved) to guide the OOD synthesis for training. The results are shown below:
>
> | Mask Ratio | FPR95↓ | AUROC↑ |
> |:----------:|:------:|:------:|
> |    1/16    |  10.31 |  **97.82** |
> |    2/16    |  **10.27** |  97.81 |
> |    4/16    |  10.47 |  97.78 |
> |    8/16    |  11.02 |  97.73 |
> |    16/16   |  11.70 |  97.62 |
>
> It indicates that masking 1~2 words/tokens is **effective enough** for perturbation guidance. Masking 4 or more words even leads to performance degradation, which means severely perturbed contexts may choose the noisy OOD candidates (e.g., random noise), making the learned spurious context unable to capture the true spurious OOD samples and further describe the category boundary.
>
> We will add the ablation studies and discussions to the manuscript.
>
> > Q2: Reciprocal Points Learning (RPL)$^{[1]}$ explores a similar idea of spurious context, which should be cited and discussed.
>
> Thanks for your constructive suggestion. We will add the discussion and cite it in the manuscript.
>
> The similarity between our method and Reciprocal Points Learning (RPL)$^{[1]}$ and its sequel ARPL $^{[2]}$ is that we all explicitly learn the spurious contexts / reciprocal points beyond the ID category to deal with unknown samples in the open-world. As follows, we briefly discuss the difference from two main aspects:
> 1. The **motivation** between our method and RPL/ARPL is different. RPL/ARPL aim at modeling the whole complementary feature space against a certain ID category for a visual classifier. On the contrary, our paper focuses on leveraging the pretrained vision-language models to capture the spurious OOD samples surrounding a certain ID category, and hierarchically describe the category boundary.
> 2. The **approach** between our method and RPL/ARPL is different. RPL/ARPL are essentially learning a more compact in-distribution feature space by the constraints from reciprocal points, which are more similar to NPOS $^{[3]}$ and VOS $^{[4]}$. In contrast, our method freezes the model's parameters to maintain the generalized feature space, and learns the spurious contexts in a specially-designed prompt-tuning way.
>
>
> [1] Guangyao Chen. Learning Open Set Network with Discriminative Reciprocal Points, ECCV 2020.
>
> [2] Guangyao Chen. Adversarial Reciprocal Points Learning for Open Set Recognition, TPAMI 2021.
>
> [3] Leitian Tao. Non-Parametric Outlier Synthesis, ICLR 2023.
>
> [4] Xuefeng Du. VOS: Learning What You Don't Know by Virtual Outlier Synthesis, ICLR 2022.
>
> > Q3: I don't quite understand why the perceptual context is called hierarchical.
>
> We are sorry for that. Actually, the perceptual context together with the spurious contexts are called hierarchical contexts.
>
> In our paper, the term "hierarchical" does NOT mean a taxonomic hierarchy, e.g., a fish node/category with two leaves/subcategories (goldfish and lionfish) in the ImageNet hierarchy. Instead, our "hierarchical" term refers to a mathematical/logical hierarchy, that is, first using the perceptual context to classify the samples into different categories (different colors in Fig.1 (a)), then using the spurious context for each category to further determine whether the sample truly belongs to this category or just comes from out-of-distribution (dark regions in Fig.1 (b)).
>
> In the manuscript, we will make a more rigorous definition of the term "hierarchical" with more appropriate descriptions.

---

> > ### Comment · Reviewer_oU43 · 2023-08-16
> > **Thanks for the response**
> >
> > I would like to thank the author for their rebuttal.
> > My concerns are largely resolved.
> > I would strongly recommend paraphrasing the usage of the term hierarchical, as I see multiple reviewers raising this question.

---

> > > ### Author Response · Authors · 2023-08-17
> > > **Thanks for the constructive feedback**
> > >
> > > We sincerely thank the reviewer again for evaluating our work and providing constructive feedback.
> > >
> > > We did neglect rigorous definitions of the term "hierarchical" in the initial manuscript. In the revised version, we will paraphrase the usage of "hierarchical" in the introduction/abstract section.
> > >
> > > ---
> > >
> > > Best regards,
> > >
> > > Authors

---

### Official Review · Reviewer_rNwN · 2023-07-07

**Soundness:** 3 good
**Presentation:** 2 fair
**Contribution:** 2 fair
**Rating:** 6
**Confidence:** 4

**Summary:**

This paper proposes a framework for detecting out-of-distribution (OOD) samples using hierarchical description - perceptual and spurious contexts. Authors consider a category-extensible setup where categories can merge hierarchically. The proposed approach is evaluated by considering ImageNet as the in-distribution data and iNaturalist, SUN, Places, and Texture as OOD datasets.

**Strengths:**

1. Considering image-text models to learn precise class boundaries is interesting and effective.

2. Authors performed extensive experiments on relevant benchmarks.

3. Results are state of the art.

**Weaknesses:**

1. It is not clear from the introduction what the hierarchical contexts are and how they affect OOD detection in a category-extensible way. Precisely, how do these contexts influence Fig. 1(a) to get updated to Fig. 1(b)?

2. Authors talk about learning precise class boundaries. This is perhaps more suitable for novel class detection i.e., open-set learning than OOD detection. For example, the same class 'Car' can be in-distribution in a sunny environment and OOD in a rainy environment. How does the proposed approach address this scenario?

3. Line 150-152: how does the combination of the contexts implemented? It is not clear from the description.

4. section 3.2 and Fig. 3: how are the spurious samples generated and perturbed? Are the perturbations always random or informed by text cues?

**Questions:**

Please address the questions in the 'Weaknesses' section.

**Limitations:**

Yes

---

> ### Author Rebuttal · Authors · 2023-08-09
>
> We are glad that the reviewer considers our method to be interesting and effective.
>
> However, we are sorry that you get confused with the mechanism and procedure of our method, due to our imperfect presentation. With the illustration figures in our uploaded PDF file, we hope the following responses can successfully address your concerns.
>
> > Q1: It is not clear from the introduction what the hierarchical contexts are and how they affect OOD detection in a category-extensible way. Precisely, how do these contexts influence Fig. 1(a) to get updated to Fig. 1(b)?
>
> In our paper, the term "hierarchical" does NOT mean a taxonomic hierarchy, e.g., a fish node/category with two leave nodes/subcategories (goldfish and lionfish) in the ImageNet hierarchy. Instead, our "hierarchical" term refers to a mathematical/logical hierarchy, that is, first using the perceptual context to classify the samples into different categories (different colors in Fig.1 (a)), then using the spurious context for each category to further determine whether the sample truly belongs to this category or just comes from out-of-distribution (dark regions in Fig.1 (b)).
>
> As illustrated in Fig.4 (c-d), the learned hierarchical contexts (i.e., perceptual and spurious contexts) are essential for distinguishing base/novel classes and ID/OOD samples in the category-extensible way. Besides, we have revised the figure and caption in the uploaded PDF file (**Figure A3**), which may be more clear and comprehensible.
>
> > Q2: Authors talk about learning precise class boundaries. This is perhaps more suitable for novel class detection i.e., open-set learning than OOD detection. For example, the same class 'Car' can be in-distribution in a sunny environment and OOD in a rainy environment. How does the proposed approach address this scenario?
>
> Thanks for pointing out this problem. Indeed, in the early stage, the task definitions were confusing, but nowadays out-of-distribution detection (OOD) and open-set recognition (OSR) tend to be a unified scenario $^{[1][2][3]}$. Here is a brief summary of the two tasks.
>
> Traditionally, the OSR task aims at classifying samples from predefined classes and identifying the rest samples as unknown, while the initial OOD task only focuses on detecting and rejecting the unknown samples. However, modern OOD methods simultaneously take in-distribution classification and out-of-distribution detection into account. Hence, in today's community, OOD and OSR are exactly performing the same task, and the only difference lies in the evaluation settings $^{[3]}$. So you may simply view OOD and OSR the same scenario, and the precise class boundaries play the key role.
>
> Besides, in a general OOD detection setting, when defining the 'car' as an in-distribution category, a car in sunny or rainy environments will possess a large intra-class variance. This is indeed a challenging problem, and our solution is freezing the encoders of CLIP to maintain the generalized representation capability to overcome the variance.
>
> If we misunderstood or have not addressed your concern, please feel free to inform us anytime.
>
>
> [1] Yang Jingkang. Generalized Out-of-Distribution Detection: A Survey. ArXiv, 2021.
>
> [2] Yang Jingkang. OpenOOD: Benchmarking Generalized Out-of-Distribution Detection. NIPS 2022.
>
> [3] Jun Cen. The Devil is in the Wrongly-classified Samples, ICLR 2023.
>
> > Q3: Line 150-152: how does the combination of the contexts implemented?
>
> As for the combination of the perceptual and spurious contexts, the methodological mechanism is as we discussed in Q1, and the mathematical formulation please refers to Eq.(1-3) (for training) and Eq.(5) (for inference). For example, in Eq.(5), we use the perceptual context $w_k^p$ to compute the initial image-text similarity $s_k$ with the $k$-th ID category. Then the spurious contexts $w_k^s$ is adopted to calculate the regularization item $\gamma_k$, which presents the probability that this input image truly belongs to the $k$-th ID category or comes from out-of-distribution. Finally, we combine the two contexts by multiplying the initial similarity $s_k$ by the regularization item $\gamma_k$, to derive the final score $r_k=s_k \times \gamma_k$ for ID classification and OOD detection.
>
> Besides, we have illustrated the whole procedure in **Figure A2** in the uploaded PDF file. If there is still something unclear, please let us know and we are happy to answer your further questions at our best effort.
>
> > Q4: section 3.2 and Fig. 3: how are the spurious samples generated and perturbed? Are the perturbations always random or informed by text cues?
>
> We first randomly generate a set of candidate samples in the image feature space, and then use the perturbed text features to select valid samples. The selected samples are so-called spurious samples. And indeed, the perturbations are always random, because during the training iterations, randomly perturbing is effective enough to guide the OOD sample generation, as illustrated in our supplementary material.
>
> For clear illustration, we provide the **visualizations** of the initial OOD **sample generation**, our **perturbation guidance**, and further **training goals** in **Figure A1** in the uploaded PDF file. We hope it can help understand our method.

---

### Official Review · Reviewer_ufZG · 2023-07-10

**Soundness:** 2 fair
**Presentation:** 2 fair
**Contribution:** 3 good
**Rating:** 6
**Confidence:** 3

**Summary:**

This paper presents a solution for the task of out-of-distribution detection with a hierarchical context. Specifically, it introduces the concepts of spurious context as negative descriptions to learn the distribution of unseen categories implicitly. When equipped with perceptual context, it can effectively detect OOD samples. Experimental results verify its robustness and effectiveness on several benchmarks.

**Strengths:**

- The proposed method is simple yet effective. Instead of finding a negative context for all categories, it proposes to adopt one for each category. Besides, the approach proposed to find negative samples is novel and intuitive.

- The writing structure is clear and easy to follow. However, I find it hard to have an initial guess for the meanings of "perceptual context" and "spurious" context in the abstract section.

- The experimental results are strong and solid. Besides, the ablation study in Tab.4 verifies the effectiveness of both proposed modules.

**Weaknesses:**

- The masking ratio in section 3.2, as a very important factor, hasn't been studied. A too-small ratio may lead to some false negative candidates while too-large ratios can neglect false positive candidates. From this perspective, this hyper-parameter can be very sensitive, thus influencing performance vastly.

- I just doubt whether a single vector w^p_k for a spurious can model the complex category boundary. Modeling the interior region of a category is intuitive as in the ideal case it can model a quasi-hypersphere. But for OOD samples surrounding a category, the manifold can be more complex.

- Fig.4 (a-b) as the main visualization, is a bit confusing. It's hard to tell what it aims to show.

**Questions:**

What is the masking ratio in Pertubantion Guidance? And is the performance sensitive to it?

Where will the spurious context locate? How about plotting them in Fig 4(a-b)?
Although there has been an illustrative plotting in Fig.1, what would it be like in TSNE visualization?

**Limitations:**

No potential negative societal impact

---

> ### Author Rebuttal · Authors · 2023-08-09
>
> We are really encouraged that the reviewer recognizes our method to be simple, novel, intuitive, and effective.
>
> We thank the valuable comments and insightful suggestions, and we hope our detailed responses below can address your concerns.
>
> > Q0: However, I find it hard to have an initial guess for the meanings of "perceptual context" and "spurious context" in the abstract section.
>
> Thank for pointing out this. We did miss a concise and precise description in the abstract section. Specifically, we are planning to revise corresponding sentences into:
>
> "Perceptual contexts perceive the inter-category difference (e.g., cats vs apples) for current classification tasks, while spurious contexts further identify spurious (similar but exactly not) OOD samples for each single category (e.g., cats vs panthers, apples vs peaches). "
>
> We hope such a revision can help readers get into our paper faster.
>
> > Q1: The masking ratio in section 3.2 hasn't been studied.
>
> Thanks for the essential suggestion! In fact, in the supplementary materials we have demonstrated that perturbing (masking is one of the perturbing approaches) one word/token in the perceptual context is **effective** to guide the OOD syntheses in section 3.2. Moreover, we have conducted a series of ablations to study how the masking ratio influences the final OOD detection performance. Taking ImageNet-100 as the ID dataset, we learn 16 words/tokens for each category, and randomly perturb 1/16 to 16/16 words (the *classname* is always preserved) to guide the OOD synthesis for training. The results are shown below:
>
> | Mask Ratio | FPR95↓ | AUROC↑ |
> |:----------:|:------:|:------:|
> |    1/16    |  10.31 |  **97.82** |
> |    2/16    |  **10.27** |  97.81 |
> |    4/16    |  10.47 |  97.78 |
> |    8/16    |  11.02 |  97.73 |
> |    16/16   |  11.70 |  97.62 |
>
> It indicates that masking 1~2 words/tokens is **effective enough** for perturbation guidance. Masking 4 or more words even leads to performance degradation, which means severely perturbed contexts may choose the noisy OOD candidates (e.g., random noise), making the learned spurious context unable to capture the true spurious OOD samples and further describe the category boundary.
>
> We will add the ablation studies and discussions to the manuscript.
>
> > Q2: Whether a single vector $w_k^p$ for a spurious can model the complex category boundary.
>
> We really appreciate the comment! We recognize that for each ID category (e.g., cat), the spurious OOD samples are diverse (e.g., panthers, lions, etc). Thus, we aim to only describe the spurious sample surrounding the ID category, rather than the whole OOD space (which is too complicated). According to common sense, it is intuitive to use more spurious contexts ($w_k^p$) to describe better category boundaries. To verify it, we have tested the number of spurious contexts for each ID category (taking ImageNet-100 as the ID dataset), and the results are shown below:
>
> | Spurious Context Number | FPR95↓ | AUROC↑ |
> |:-----------------------:|:------:|:------:|
> |            1            |  10.31 |  97.82 |
> |            2            |  **10.21** |  97.86 |
> |            4            |  10.27 |  **97.88** |
> |            8            |  10.25 |  97.86 |
>
> It implies using more spurious contexts only leads to 0.1% gain on performance. The reason may be that the learned 2 or more spurious contexts are too redundant without any constraints. To alleviate this problem, we simply add an orthogonal constraint (making the similarities between each two spurious contexts close to zero), and the OOD detection performance is significantly boosted:
>
>
> | Spurious Context Number | FPR95↓ | AUROC↑ |
> |:-----------------------:|:------:|:------:|
> |            1            |  10.31 |  97.82 |
> |         2 + orth        |  10.17 |  97.86 |
> |         4 + orth        |   9.89 |  **97.89** |
> |         8 + orth        |   **9.76** |  97.84 |
>
> Therefore, how to effectively and efficiently leverage more spurious contexts to better describe the category boundary deserves further exploration, and we view it as our future work.
>
> We will update the experiments and discussions.
>
> > Q3: Fig.4 (a-b) as the main visualization, is a bit confusing. It's hard to tell what it aims to show. Where will the spurious context locate? How about plotting them in Fig 4(a-b)? Although there has been an illustrative plotting in Fig.1, what would it be like in TSNE visualization?
>
> Fig.4 highlights the advantages of our method against competitors, including:
> 1. Fig.4 (a) indicates finetuning the model's encoder will damage the generalized feature space, making the unseen OOD images indistinguishable from seen ID images. On the contrary, Fig.4 (b) shows our method maintains the feature-level separability by freezing the whole encoders.
> 2. Fig.4 (c-d) implies our learned spurious context (blue star in (d)) assists perceptual context (yellow star) in better distinguishing the ID and OOD samples.
>
>
> We have revised the figure and caption in the uploaded PDF file (**Figure A3**), which may be more clear and comprehensible. Besides, visualizing the spurious contexts in Fig.4 (a-b) will make the pattern too complicated to highlight the necessity to freeze the generalized feature space. Thus, we turn to visualize the spurious context in Fig.4 (d) for clarity.

---

### Author Rebuttal · Authors · 2023-08-09

We thank all the reviewers for their time, insightful suggestions, and valuable comments. We are encouraged to see **ALL** reviewers find our method **interesting** and **effective** (ufZG, rNwN, oU43, R97w, 194m), **comprehensive** experiments and **promising** results (ufZG, rNwN, oU43, R97w, 194m), and **transferable** to other areas involving large-scale vision-language models (R97w).

However, reviewers still have some extra concerns, which mainly focus on the following aspects:
* insufficient presentation on the term definition, method operation, and novelty clarification;
* lack of additional experiments and ablation studies;
* extension and combination with zero-shot applications, future explorations, etc.

We respond to each reviewer's comments in detail below, and here is a brief summary of the main discussions and experiments:
* more precise explanations of our proposed perceptual context, spurious context, hierarchical mechanism, etc;
* walkthrough examples to illustrate the perturbation process, visualize the generated samples, and demonstrate the full training procedure (in the **uploaded PDF file**);
* extensive ablation studies and discussions on the mask ratio for Perturbation Guidance, the number of learnable spurious contexts, as well as comparison and discussion with relevant works like RPL and CoCoOp;
* new insights and validations for zero-shot applications, including the combination with large language models (like GPT).

We thank the reviewers' valuable suggestions again, and we believe those make our paper much stronger.

---

### Decision · Program_Chairs · 2023-09-21

**Decision:**

Accept (poster)

**Comment:**

This paper was reviewed by five experts in the field. The authors' rebuttal resolved most of the concerns. Reviewers liked the overall proposed method and comprehensive experiments.

The AC agrees with the reviewers' assessments and believes the paper provides a simple yet effective method for OOD detection using vision-language models. The decision is to recommend the paper for acceptance. The reviewers did raise some valuable suggestions in the discussion that should be incorporated in the final camera-ready version of the paper (e.g., improving paper presentation and clarity). The authors are encouraged to make the necessary changes to the best of their ability.

Why not spotlight / oral: The current paper still requires further refinement in its final version to incorporate reviewers' suggestions.